# NEURAL MODULAR PHYSICS
# FOR ELASTIC SIMULATION

## ABSTRACT

Learning-based methods have made significant progress in physics simulation, typically approximating dynamics with a monolithic end-to-end optimized neural network. Although these models offer an effective way to simulation, they may lose essential features compared to traditional numerical simulators, such as physical interpretability and reliability. Drawing inspiration from classical simulators that operate in a modular fashion, this paper presents *Neural Modular Physics* (NMP) for elastic simulation, which combines the approximation capacity of neural networks with the physical reliability of traditional simulators. Beyond the previous monolithic learning paradigm, NMP enables direct supervision of intermediate quantities and physical constraints by decomposing elastic dynamics into physically meaningful neural modules connected through intermediate physical quantities. With a specialized architecture and training strategy, our method transforms the numerical computation flow into a modular neural simulator, achieving improved physical consistency and generalizability. Experimentally, NMP demonstrates superior generalization to unseen initial conditions and resolutions, stable long-horizon simulation, better preservation of physical properties compared to other neural simulators, and greater feasibility in scenarios with unknown underlying dynamics than traditional simulators.

## 1 INTRODUCTION

Learning-based simulators have emerged as a powerful approach to modeling complex physical systems, where neural networks are usually end-to-end optimized from data as a whole to approximate object dynamics. Representative neural simulators include neural operators (Lu et al., 2019; Li et al., 2021; Kovachki et al., 2023; Wu et al., 2024b) and Graph Neural Network (GNN) simulators (Sanchez-Gonzalez et al., 2018; 2020; Pfaff et al., 2021; Lam et al., 2023; Halimi et al., 2023), which promise fast, flexible and differentiable simulations, showing promising results in fluid dynamics (Tompson et al., 2017), material science (Friederich et al., 2021), and beyond. However, the above neural simulators typically suffer from key limitations regarding physical soundness. Specifically, neural simulators often behave as an indivisible function, where the only accessible physics quantity is the model output, thereby leaving no access to intermediate physical quantities (e.g., internal forces, stress) essential for simulation. Such opacity seriously damages simulator interpretability and also makes it difficult to enforce known physical constraints. Although some physics-informed neural networks (Raissi et al., 2019; Cuomo et al., 2022; Guo et al., 2020; Wang & Zhong, 2024; Wang et al., 2021; Stuyck et al., 2025) can explicitly optimize the model with physics equations, they usually suffer from serious training difficulties (Daw et al., 2023; Wu et al., 2024a) and are mainly limited to simple equations and scenarios (Wang et al., 2023; 2024), which are hard to support the simulation of complex long-horizon dynamics focused on in this paper.

Note that a physical trajectory is far beyond a sequence of states following a vague data distribution. Taking the dynamics of an elastic soft body as an example, the dynamics at each step are governed by global conservation laws, local geometric constraints, and material-specific constitutive relations, which together confine motion to a narrow subset of the mathematically possible state space, necessitating the physical soundness of simulators. Actually, unlike recent neural simulators, we notice that all the above-mentioned physical laws are carefully maintained in traditional numerical simulators, such as finite element methods (Ŝolín, 2005). In general, these classical simulators work modularly, which decomposes complex physical dynamics into several sub-modules and defines the interaction

among intermediate quantities based on physical prior knowledge. The modular computation flow in classical simulators not only decomposes the complex dynamics into easier sub-processes but also offers a flexible interface to constrain physical quantities for strict physical soundness. This observation motivates introducing *modularity* of classical simulators into learning-based simulators.

Previous researchers have explored some hybrid neural-physics simulators by substituting one of the modules in classical simulators with learnable neural networks, such as replacing discretization stencils (Bar-Sinai et al., 2019) or constitutive laws (Ma et al., 2023), which can be viewed as incidental and initial explorations of the modular motivation. However, since pure numerical methods are less adaptable in scenarios with unknown information (e.g., environment frictional forces), these hybrid approaches only achieve a compromise performance, which sacrifices scenario flexibility of data-driven neural simulators for better physical correctness. These piecemeal and unsatisfactory approaches naturally raise a question: *could we design neural simulators with a thorough modular architecture, maintaining both data-driven flexibility and physical soundness?* After a comprehensive empirical investigation, we find that the correct answer for modularization of physics simulation is far beyond simple substitution, which requires elaborate design to faithfully ensure physical alignment with numerical simulators and avoid *collapse* in complex modular architecture (Mittal et al., 2022). Here, "collapse" refers to confused module capability, a foundation issue of modular neural networks (Jarvis et al., 2023), and will cause undesirable module behavior and overall performance.

In this paper, we propose a *Neural Modular Physics* (NMP) framework that successfully embraces a modular learning philosophy for neural elastic simulation, which involves a specialized architecture and a modular physics training strategy. Specifically, our approach decomposes physical simulation into well-defined modules, replacing key components—such as constitutive models and time integration schemes—with specialized neural networks. This physically aligned modular design exposes essential physical quantities in the middle of the neural simulator, thereby enabling direct supervision of intermediate quantities and flexible interchange between neural and numerical implementations. Additionally, to avoid the potential collapse of the modular architecture, NMP employs a two-stage training strategy, where components are first trained independently, supervised by intermediate physics quantities of numerical simulators, before being fine-tuned together, enabling the *specialization* and stricter physical alignment of each module. Furthermore, benefiting from modular design, NMP achieves favorable flexibility in enforcing physical constraints. By utilizing exposed intermediate physical quantities, such as the deformation gradient of an elastic body, NMP enables more detailed physics constraints, such as local volume preservation, which cannot be accomplished in previous end-to-end neural simulators, demonstrating the potential of this new modular learning paradigm. We demonstrate these advantages through comprehensive experiments across diverse elastic simulation tasks. Our contributions can be summarized as follows:

- We propose and investigate a new thorough modular neural framework *Neural Modular Physics* for elastic simulation. Every classical sub-system can be replaced by a neural module with well-defined physical interfaces, enabling direct supervision of intermediate quantities, interchangeability with physics-based modules and flexible physical constraints.

- Specialized architecture and modular physics training strategy are presented to ensure better physical alignment to numerical methods and avoid collapse in complex modular architectures, which successfully guarantees the stable training and module specialization.

- NMP demonstrates strong generalization to unseen initial conditions and resolutions, delivering accurate long-horizon rollouts that outperform advanced neural simulators and showing better flexibility in scenarios with unknown information than hybrid simulators.

## 2 RELATED WORKS

### 2.1 NEURAL SIMULATORS

As an advanced technique, deep models have been widely explored in physical simulation (Pfaff et al., 2021; Ma et al., 2023; Wu et al., 2024b). Previous neural simulators can be roughly categorized into the following two paradigms: end-to-end neural and hybrid neural-physics simulators.

In end-to-end neural simulators, neural networks with diverse architectures have been used as a whole approximator to mimic physical dynamics supervised from data. For example, Convolutional Neural Networks (Guo et al., 2016; Tompson et al., 2017; Afshar et al., 2019; Liu et al., 2023) have

been explored to capture spatial patterns in Eulerian physical systems. To handle complex geometries, Graph Neural Networks (GNNs)-based methods, such as MeshGraphNet and others (Sanchez-Gonzalez et al., 2020; Pfaff et al., 2021; Lam et al., 2023; Sanchez-Gonzalez et al., 2018; Halimi et al., 2023), leverage message passing to model complex interactions between system elements. Recently, neural operators (Li et al., 2021; Lu et al., 2019; Kovachki et al., 2023; Wu et al., 2024b) are presented to approximate physics in the function space, providing a framework to predict physical processes across different resolutions and boundary conditions. While promising, these approaches rely on end-to-end architectures that model the whole physical system with a monolithic network, which makes it hard to guarantee physical laws. In contrast, we disentangle the complex physics in a modular architecture, enabling better physical interpretability and more flexible physical constraints.

As for hybrid neural-physics methods, these approaches embed learning at targeted stages of a classical solver, capturing effects that analytic models miss while preserving a physical backbone. Examples include learned data-driven discretization stencils (Bar-Sinai et al., 2019) and learned residual dynamics on top of analytical dynamics to account for the unmodeled part (Yin et al., 2021). In elastic simulation, NCLaw (Ma et al., 2023) learns the constitutive law. However, existing hybrids typically replace only part of the components and leave the rest of the pipeline to classical solvers, resulting in compromised performance in scenario flexibility and physics consistency. We advance this line by modularizing the entire elastic simulation pipeline. Thorough modularization delivers the expressive power of learning while preserving the organizing principles of classical mechanics.

## 2.2 Neural Modular Networks

Neural modular networks promote specialization and interpretability by assigning distinct sub-tasks to separate sub-nets. Andreas et al. (2016) initialized this modularity idea by dynamically composing task-specific graphs for language reasoning, which is subsequently used for visual-question answering and program-synthesis (Kirsch et al., 2018; Lu et al., 1995). Although the modularity of neural networks has been shown to be beneficial in wide applications, its effectiveness inherently depends on the specialization of the learned modules (Mittal et al., 2022; Jarvis et al., 2023) and has not been well explored in physics simulators. In this paper, we observe the internal modularity in numerical simulators, motivating us to present Neural Modular Physics to modularize the physics computation process, along with specialized architecture and training strategy to avoid collapse.

## 3 Neural Modular Physics

As mentioned earlier, this paper attempts to construct a thorough modular framework for elastic simulation towards better scenario flexibility and physical soundness. To ensure physical alignment and module specialization, we present Neural Modular Physics (NMP) with specialized modular architecture and training strategy. This section will first introduce some basic knowledge and insights from classical simulators and then detail the concrete design for formalizing and training NMP.

### 3.1 Inspirations from Classical Simulators

Since module specialization is the key in neural modular networks (Mittal et al., 2022), it is usually hard but essential to decide model modules. Fortunately, unlike language or visual usages, physical simulation has a long-standing reference, that is, classical numerical simulators. Thus, we propose to follow the computation process of classical simulators, which not only provides native heterogeneous modules but also leaves physically meaningful interfaces for subsequent optimization.

Specifically, the motion of a deformable object obeys Newton's second law $\mathbf{M}\ddot{\mathbf{x}} = \mathbf{f}_{\text{int}}(\mathbf{x}) + \mathbf{f}_{\text{ext}}$, where $\mathbf{x} \in \mathbb{R}^{3n}$ stacks the nodal positions of a mesh with $n$ vertices, $\mathbf{M}$ is the (lumped) mass matrix, $\mathbf{f}_{\text{int}}$ denotes internal elastic forces, and $\mathbf{f}_{\text{ext}}$ collects external loads such as gravity or contact. A standard finite-element method (FEM) based on discretization under tetrahedral or hexahedral elements involves the following computation: (i) first computes the deformation gradient $\mathbf{F}$ inside each element from nodal displacements, and then assembles element forces into the global system, (ii) along with a time integration to simulate dynamics. The above-described computation obviously derives two decoupled modules: spatial force computation (known as the Constitutive law) and time integration. Inspired by this observation, NMP modularizes the elastic simulation into *neural constitutive module* and *neural integration module*, which will be detailed in the next section.

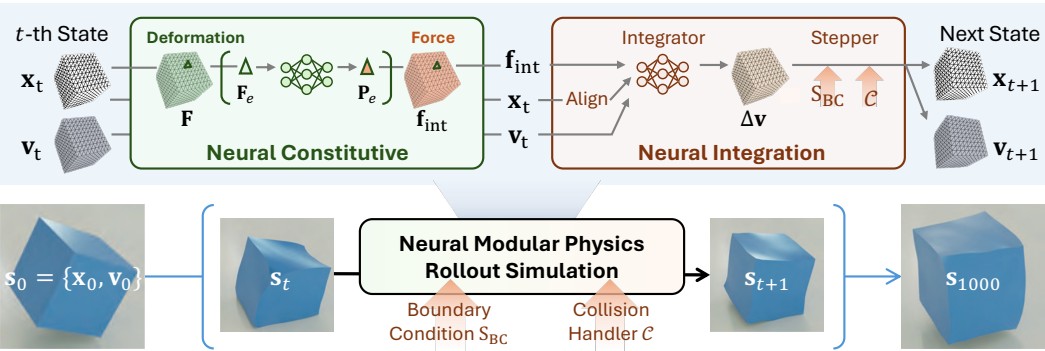

Figure 1: Neural Modular Physics (NMP) modularizes elastic physics simulation into two parts: neural constitutive module to compute internal forces $\mathbf{f}_{\text{int}}$, and neural integration module to evolve system state $\mathbf{s}_t = \{\mathbf{x}_t, \mathbf{v}_t\}$ and handle traditional boundary conditions $\mathbf{S}_{BC}$ and collision $\mathcal{C}$.

## 3.2 ELASTIC PHYSICS MODULARIZATION

Drawing inspiration from classical simulators, we factorize the traditional computation process into two successive neural modules, each aligned with a well-defined physical sub-process (Fig. 1).

Specifically, given the system state $\mathbf{s}_t$ with vertex position $\mathbf{x}_t$ and velocity $\mathbf{v}_t$, neural constitutive module maps deformation gradient $\mathbf{F}$ computed from nodal displacements to internal forces $\mathbf{f}_{\text{int}}$, while neural integration evolves $\mathbf{s}_t$ and completes each step with boundary condition enforcement ($\mathbf{S}_{BC}$) and collision handling ($\mathcal{C}$), maintaining traditional simulator's computational flow (Alg. 1). Here are the details of each module.

**Neural constitutive module** In FEM, internal forces $\mathbf{f}_{\text{int}}$ are determined by the material's constitutive law through the following process. First, the strain energy density function $\Psi(\mathbf{F})$ defines the material's response to deformation, where $\mathbf{F} = \frac{\partial \mathbf{x}}{\partial \mathbf{X}}$ is the deformation gradient measuring

---

**Algorithm 1** Neural Modular Physics

**Input:** $\mathbf{s}_t = \{\mathbf{x}_t, \mathbf{v}_t\}, \mathbf{S}_{BC}, \mathcal{C}$
**Output:** $\mathbf{s}_{t+1} = \{\mathbf{x}_{t+1}, \mathbf{v}_{t+1}\}$
*// Neural constitutive module*
  **foreach** *element e* **do**
    Compute deformation gradient $\mathbf{F}_e$ from $\mathbf{x}_t$
    Predict stress $\mathbf{P}_e = f_\theta(\mathbf{F}_e)$
    Compute element forces $\mathbf{f}_{\text{int}}^e = \mathbf{B}_e^\top \mathbf{P}_e V_e$
Assemble global force $\mathbf{f}_{\text{int}} = \sum_e \mathbf{f}_{\text{int}}^e$
*// Neural integration module*
$\Delta\mathbf{v} = g_\phi(\mathbf{x}_t, \mathbf{v}_t, \mathbf{f}_{\text{int}})$
$\mathbf{v}_{t+1} = \mathbf{v}_t + \Delta\mathbf{v}$
$\mathbf{x}_{t+1} = \mathbf{x}_t + h\mathbf{v}_{t+1}$
$\mathbf{x}_{t+1}, \mathbf{v}_{t+1} = \mathcal{C}(\mathbf{S}_{BC}(\mathbf{x}_{t+1}, \mathbf{v}_{t+1}))$

---

local geometric distortion from reference coordinates $\mathbf{X}$ to deformed coordinates $\mathbf{x}$. Then, the first Piola-Kirchhoff stress $\mathbf{P}$ is computed as $\mathbf{P} = \frac{\partial\Psi(\mathbf{F})}{\partial\mathbf{F}}$, representing the material's resistance to deformation. Internal forces are calculated by adding up all elements: $\mathbf{f}_{\text{int}} = \sum_e \mathbf{B}_e^\top \mathbf{P} V_e$, where $\mathbf{B}_e$ is the strain-displacement matrix for element $e$, $V_e$ is the element's volume in the reference configuration.

In this module, the classical computation flow is maintained except that the computation of the stress term $\mathbf{P}$ is replaced by a learnable neural network $f_\theta$, since the material response is usually hard to measure in real-world scenarios, where the data-driven paradigm is more flexible. Here, $f_\theta$ is configured as a special rotation equivariant architecture following Ma et al. (2023). Concretely, given $\mathbf{F} = \mathbf{U\Sigma V}^\top$, we feed the rotation-invariant tuple $(\mathbf{\Sigma}, \mathbf{F}^\top \mathbf{F}, \det\mathbf{F})$ into a two-layer MLP and then rotate the output back with $\mathbf{R} = \mathbf{UV}^\top$, yielding $\mathbf{P} = \mathbf{R} f_\theta(\cdot)$. The above-described modularization maintains FEM's element-wise computational structure while leveraging learning to capture complex material behaviors that traditional models may fail to capture in real-world applications.

Notably, the parameterization of $f_\theta$ adopted from Ma et al. (2023) is only a part of our thorough modular framework, which pursues a fully modularized neural simulator and adopts a more complete methodology to tackle newly emerged difficulties in physics modularization and model optimization.

**Neural integration module** In the classical FEM, the equations of motion can be integrated over time using various schemes, with the most common being explicit and implicit Euler:

$$\mathbf{x}_{t+1} = \mathbf{x}_t + h\mathbf{v}_{t+\alpha}, \; \mathbf{v}_{t+1} = \mathbf{v}_t + h\mathbf{M}^{-1}\left(\mathbf{f}_{\text{int}}(\mathbf{x}_{t+\alpha}) + \mathbf{f}_{\text{ext}}\right), \quad (1)$$

where $h$ is the timestep and the choice of $\alpha$ selects an integration scheme: $\alpha = 0$ (explicit Euler) and $\alpha = 1$ (implicit Euler). $\mathbf{M}$ denotes the (lumped) mass matrix and $\mathbf{f}_{\text{ext}}$ represents the external force,

Figure 2: Modular physics training strategy and inference scheme. A two-stage training paradigm is employed to optimize neural modules, where the first-stage separate training can effectively avoid module collapse and the second joint finetuning can further leverage the physical constraints.

which are both hard to access in real-world applications. For example, it is hard to decide the mass matrix $\mathbf{M}$ since the object can be hollow or solid and the environment's frictional forces depend on the contact material. This motivates us to compute $\mathbf{M}$ and $\mathbf{f}_{\text{ext}}$ related terms in a learnable way.

Specifically, the neural integration module preserves the same interface as an implicit FEM, but approximates velocity increment $\Delta\mathbf{v}$ ($\mathbf{M}$, $\mathbf{f}_{\text{ext}}$-related term in Eq. 1) with a learnable neural network:

$$\Delta\mathbf{v} = g_\phi(\mathbf{x}_t, \mathbf{v}_t, \mathbf{f}_{\text{int}}) \quad \text{with} \quad \mathbf{v}_{t+1} = \mathbf{v}_t + \Delta\mathbf{v}, \ \mathbf{x}_{t+1} = \mathbf{x}_t + h\,\mathbf{v}_{t+1}. \tag{2}$$

For the configuration of $g_\phi$, a major obstacle is that the raw coordinates $\mathbf{x}_t$ vary by global translation and rotation between scenes and simulation steps. We therefore prepend a learnable *canonicalisation* module: a two-layer T-Net (Qi et al., 2017) predicts a $3\times3$ linear map $\mathbf{T}$ that aligns the deformed mesh to a learned reference frame, $\tilde{\mathbf{x}}_t = \mathbf{T}(\mathbf{x}_t - \bar{\mathbf{x}}_t)$, where $\bar{\mathbf{x}}_t$ is the mesh centroid. Then, we embed aligned positions, original positions, velocities and internal forces of each vertex with MLP layers; concatenating these embeddings with the raw inputs yields a compact feature vector describing the mesh state. Afterwards, another MLP then maps the feature to the velocity increment $\Delta\mathbf{v}\in\mathbb{R}^3$.

Beyond elaborative physical alignment, the above modularization has several favorable properties. First, it exposes intermediate quantities at module interfaces, which can be directly supervised by intermediate quantities of FEM during training, further boosting the model interpretability and module specialization. Second, neural modules maintain the same interface as traditional simulators, allowing interchange with their traditional physics-based counterparts for flexible deployment (Fig. 2b).

### 3.3 Modular Physics Training

As demonstrated by Mittal et al. (2022), a modular architecture is not enough to guarantee that the whole model works modularly; the training strategy is also essential. Therefore, as presented in Fig. 2, we present a two-stage training strategy to optimize the predefined neural modules.

**Separate training** In the first phase, we train the neural constitutive module and neural integration module independently, along with the differentiable simulation counterpart for the other component. Specifically, for each timestep, the supervision can be formalized as follows:

$$\begin{aligned}
&\text{Neural constitutive module } f_\theta: \mathcal{L}_{\text{constitutive}} = \|\mathbf{f}_{\text{int}} - \mathbf{f}_{\text{int}}^*\|_2^2 + \|\mathbf{x}_{\text{FEM-Integration}} - \mathbf{x}^*\|_2^2 \\
&\text{Neural integration module } g_\phi: \mathcal{L}_{\text{integration}} = \|\mathbf{v} - \mathbf{v}^*\|_2^2 + \|\mathbf{x}_{\text{FEM-constitutive}} - \mathbf{x}^*\|_2^2,
\end{aligned} \tag{3}$$

where the timestep subscript is omitted for simplification. $\cdot^*$ represents the supervision generated by FEM. Benefiting from the unique tractable property of FEM, we can also enable direct supervision for the intermediate results of neural modules, ensuring each network learns its specific role, namely, material behavior or temporal dynamics. Additionally, we also utilize the differential FEM modules along with neural counterparts to generate simulation results, where the supervision on final result $\mathbf{x}$ can serve as a regularization term to ensure that each module's output is physically meaningful.

**Joint finetuning** After separate training, we obtain two well-optimized modules. Then, we fine-tune both networks together, allowing them to adapt to each other while maintaining their physical understanding from pre-training. It is worth noting that, by leveraging the exposed intermediate physical quantities, such as deformation gradients, NMP is also able to analytically compute physical constraints, which was previously impossible in monolithic neural simulators due to their opacity.

To demonstrate our framework's flexibility in physical constraining, we further explore the incorporation of *local volume preservation* of elastic dynamics into the loss function to improve the physical consistency of simulations. Specifically, this preservation corresponds to the fact that, for nearly incompressible elastomers, the determinant of the deformation gradient, $\det(\mathbf{F})$, should stay close to 1. Thus, we additionally penalize excess deviation with a hinge loss (Gentile & Warmuth, 1998):

$$\mathcal{L}_{\text{volume}} = \sum_e \left[ \max\big( |\det(\mathbf{F}_e) - 1| - \epsilon_{\text{volume}},\, 0 \big) \right]^2, \tag{4}$$

where $e$ indexes elements and we set $\epsilon_{\text{volume}} = 0.05$ following engineering practice. During finetuning, this newly added penalty is added to the L2 loss for intermediate physical quantities and model predictions with a weight of 0.1. In experiments, this physical constraint will not drastically improve the accuracy metric but can consistently enhance the physical soundness of model predictions.

Note that the above physical constraint on local volume preservation is only an example. With our modular physics framework, it is flexible to add new physical regularization to the training process, such as the L2 distance between the elastic potential energy $\sum_e V_e\, W_e(\mathbf{F}_{e,t})$ of the neural and FEM simulators, where $V_e$ and $W_e(\cdot)$ represent rest volumes and strain-energy density respectively.

## 4 EXPERIMENTS

We construct five test environments to evaluate different aspects of elastic simulation. The first four environments are designed with complete physical information to test a model's capacity for physics learning, while the final environment introduces unknown friction to imitate a real-world challenge.

**CUBE** We simulate an elastic cube $(1,000$ vertices, $3,645$ tetrahedra) dropping and bouncing on a rigid floor under gravity. This scene tests external collision handling and general deformation behavior. The validation set tests generalization by varying initial velocity and cube orientation, examining both collision response and general deformation behavior under large dynamics.

**CUBEXL** An additional higher resolution cube CUBEXL that is made up of a grid of $22 \times 22 \times 22$ $(10,648$ vertices) is constructed to stress test the model's behavior for large-resolution meshes.

**SPOT** The little cow "Spot" (Crane et al., 2013) $(1,015$ vertices, $2,752$ tetrahedra) is constrained at both head and tail. This environment tests the handling of multiple fixed boundary points and stress propagation in complex geometry. The validation set varies initial velocities.

**BOB** An elastic duck "Bob" (Bob) $(849$ vertices, $3,108$ tetrahedra) is suspended in air with fixed head vertices. This environment tests the method's ability to handle Dirichlet boundary conditions at fixed points and complex geometry deformation. The validation dataset varies the initial velocity.

For these four environments, we use a neo-Hookean material model (Smith et al., 2018) with timestep $h = 5 \times 10^{-4}$ seconds, computed via a semi-implicit integrator. Each dataset comprises 32 training trajectories of 1,000 steps each, with randomly sampled initial velocities, plus 8 validation trajectories with unseen initial conditions. Additional details are provided in Appendix B.

**FRICTION** The final benchmark introduces unknown friction dynamics. An elastic rubber duck "Bob" (Bob) $(5,130$ vertices, $21,448$ tetrahedra) is dropped onto a rough surface and begin sliding under frictional contact dynamics. Trajectories are generated with an external Projective Dynamics simulator (Du et al., 2021), which models both hyperelastic material response and frictional contact through an implicit integration scheme. Unlike the previous environments where contact reduces to simple non-penetration constraints, this setting requires the neural network to capture rich frictional effects that are not provided explicitly. This benchmark therefore evaluates the model's ability to generalize to more realistic contact phenomena and infer missing dynamics directly from data.

All the simulation results can be found at https://sites.google.com/view/neural-modular-physics.

### 4.1 MAIN RESULTS

**Baselines** As discussed in the introduction, our proposed NMP is a thorough neural modular network. Thus, we include three representative neural simulators as baselines: GNN-based MeshGraphNet (MGN) (Pfaff et al., 2020), and advanced neural operators EGNO (Xu et al., 2024) and Transformer-based Transolver (Wu et al., 2024b). Additionally, pure and hybrid neural-physics simulators are also capable of generating simulations, which will be compared in the next section.

Table 1: We report RMSE between the model-simulated vertex positions and ground truth simulation across different rollout horizons. See Appendix C.1 for baseline visualizations.

| Method | CUBE | | | CUBEXL | | | SPOT | | | BOB | | |
|---|---|---|---|---|---|---|---|---|---|---|---|---|
| | 100 | 500 | 1000 | 100 | 500 | 1000 | 100 | 500 | 1000 | 100 | 500 | 1000 |
| Transolver (2024b) | 0.277 | 1.525 | 3.200 | 0.362 | 3.622 | 9.984 | 0.229 | 0.943 | 2.290 | 0.075 | 0.597 | 1.782 |
| EGNO (2024) | 0.503 | 2.308 | 4.492 | 0.662 | 3.288 | 6.357 | 0.370 | 1.176 | 1.217 | 0.299 | 1.414 | 2.770 |
| MeshGraphNet (2020) | 0.151 | 0.371 | 1.554 | 0.082 | 0.433 | 1.963 | 0.066 | 0.577 | 2.364 | **0.048** | 0.508 | 2.104 |
| **Ours** | **0.019** | **0.159** | **0.535** | **0.020** | **0.253** | **0.440** | **0.061** | **0.217** | **0.449** | 0.050 | **0.316** | **0.579** |

**Setups** We evaluate all methods on $100$, $500$ and $1,000$-step rollouts from unseen test trajectories across all environments, which can reveal models' long-horizon stability and accumulated physical error over various durations. Note that during training, our model supervises on sequences for at most $200$ steps, resulting in a temporal extrapolation for $500$ and $1,000$-step rollouts.

**Results** As reported in Table 1, compared with other neural simulators, NMP achieves a significantly better accuracy with over 60% error reduction in the 1,000-step rollout, highlighting the benefits of modular design in capturing long-horizon dynamics. Notably, although we have carefully tuned advanced neural operators, EGNO and Transolver, they still fail in long-term simulation. This may be that without explicit supervision on intermediate physics quantities, these monolithic neural simulators are very likely to generate meaningless results, especially under long-term rollout. See Appendix C.1 and the project website for visual comparison, Appendix C.4 for statistics of per-sequence standard deviations of Table 1, and Appendix B.4 for implementation details.

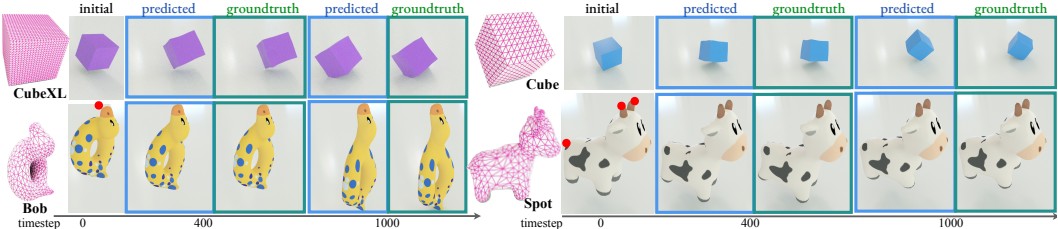

Figure 3: Visualization of NMP long-horizon simulations under random initial conditions. Each dataset shows the rest mesh and the simulation sequence at keyframes. Ours remains stable and visually matches the ground truth (FEM) even for $1,000$-step rollouts. See full videos on website.

**Visualizations** As shown in Fig. 3, although NMP was trained on sequences of at most 200 steps, it remains stable and physically plausible throughout the extended simulations. This stands in contrast to monolithic baselines, which typically diverge after a few hundred steps (Appendix C.1). These results show the efficacy of our design in supporting stable extrapolation.

**Unseen initial conditions generalization** We evaluate two unseen initial conditions—novel velocities and combined velocity + pose offsets—for BOB and CUBE (Fig. 4, left). Our modular simulator adapts to both, yielding stable, plausible $1,000$-step trajectories that closely track ground truth despite no such examples in training. Results for SPOT and CUBEXL are provided in Appendix C.3.

**Higher resolution generalization** We further test mesh-agnostic generalization by evaluating on a finer discretization than seen in training ($10,648 \rightarrow 21,952$ vertices; Fig. 4, right). Although our model is trained on one coarse resolution, our per-vertex neural modules seamlessly scale to the denser mesh, producing motions that remain stable and physically accurate.

### 4.2 COMPARISON WITH PHYSICS SIMULATORS

In this section, we elaborate on the comparison between NMP and physics simulators. Specifically, we include *(i)* a traditional physics simulator and *(ii)* the hybrid neural-physics method NCLaw (Ma et al., 2023) that augments a numerical integrator with a learned material model as baselines.

**Scenario flexibility** We evaluate on the FRICTION environment, where an elastic duck interacts with a surface through unknown frictional dynamics. Each method is tested on unseen 1,000-step

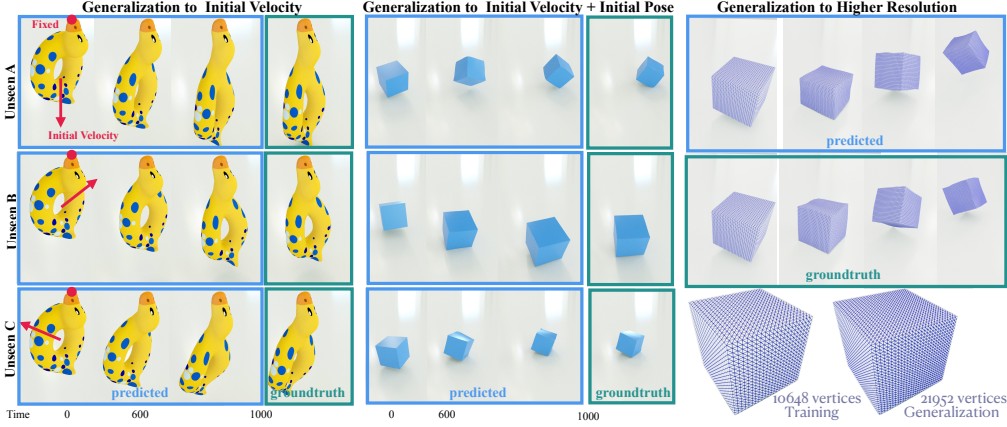

Figure 4: NMP simulations under unseen initial conditions and higher resolutions . Left to right, Ours vs. Ground Truth : unseen initial-velocity on BOB, unseen velocity+pose on CUBE, mesh-resolution generalization on CUBEXL (trained at 10k vertices, tested at 21k). See website for videos.

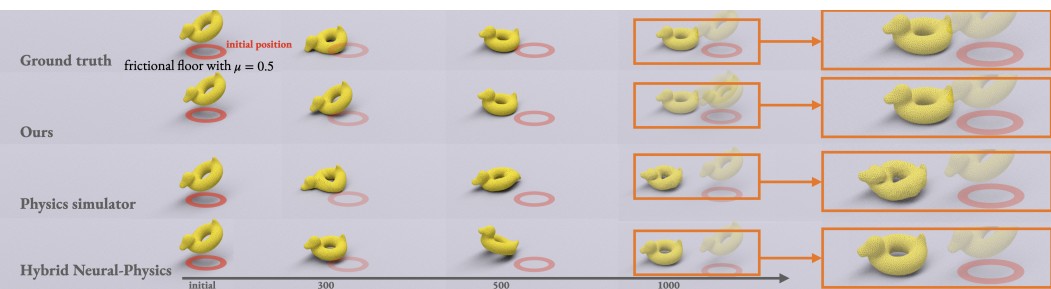

Figure 5: Simulation comparison on FRICTION environment with unknown frictional dynamics. We mark the initial position of duck-surface contact with red circles.

rollouts. Note that both *pure* and *hybrid* physics simulators without access to the underlying contact law cannot capture the unknown frictional effects. Thus, we perform system identification to obtain reasonable estimates of the physics simulator's input parameters for these two baselines. In contrast, our model can learn from data, thereby not requiring the system identification process.

As shown in Fig. 5, our approach reproduces the effect of friction: the duck stops after a short slide, rather than gliding without resistance. This indicates that our model successfully learns the unknown frictional interactions. Although NCLaw learns material response, both NCLaw and the pure physics simulator fail to capture the underlying contact dynamics. These results demonstrate that our neural simulator provides superior flexibility in scenarios with incomplete physical information. While numerical simulators perform well in complete-information settings, this benchmark underscores the unique strength of neural approaches in handling unknown dynamics.

**Efficiency comparison** In addition to scenario flexibility, another advantage of neural simulators is their efficiency (Li et al., 2021; Wu et al., 2024b). Here we also measure the running efficiency of various methods. As listed in Table 2, all neural baselines and our method exhibit significantly better efficiency than physics and neural-physics hybrid methods. This advantage comes from the simplified inference paradigm of neural simulators, which only involves the forward pass of neural networks and the tensor operations have been extremely optimized in modern devices (Paszke et al., 2019). In particular, NMP utilizes a specialized modularization architecture, which allows us to introduce only lightweight neural modules, rather than the unwieldy monolithic model. Therefore, NMP is around 40× faster than the tradi-

Table 2: Efficiency comparison among different methods on the BOB task. Inference time for 1,000-step rollout is recorded.

| Method | Time (ms) |
|---|---|
| Transolver (2024b) | 11796 |
| EGNO (2024) | 26351 |
| MeshGraphNet (2020) | 27603 |
| NCLaw (Neural-Physics) (2023) | 97121 |
| Physics Simulator | 90275 |
| **NMP (Ours)** | **2250** |

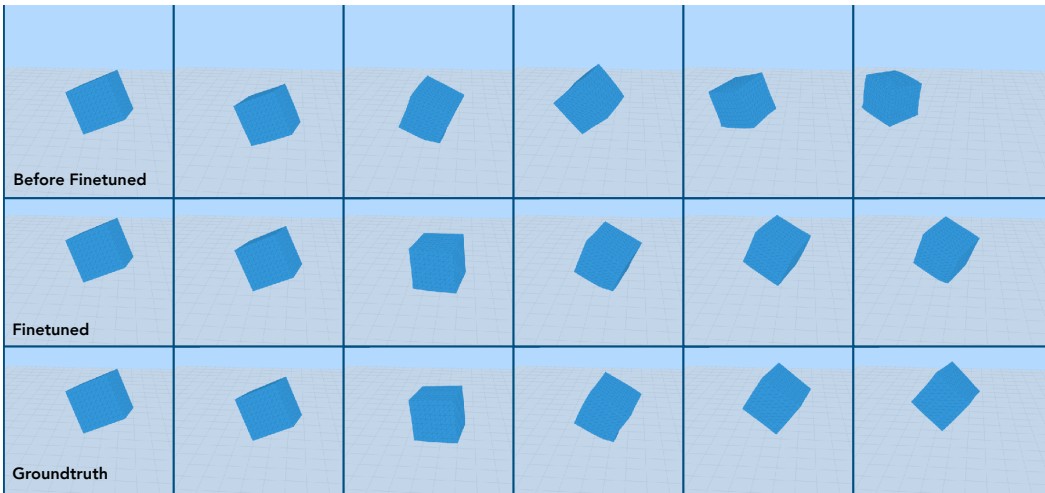

Figure 6: A proof-of-concept experiment of "sim to real" transfer. We pretrain an NMP model on the CUBE and finetune it on a scenario with unknown external force and a softer material. To mimic the real-world simulation task, only the position information is accessible during finetuning.

tional physics simulator and over $5\times$ faster than the advanced neural operator Transolver (Wu et al., 2024b), where the latter has to infer the complete unwieldy model at each step.

**Transferability** Since NMP employs a two-stage training strategy, the internal force is required (Eq. 3) during training to ensure a successful modularization. Such intermediate physics is accessible in the simulation scenario but cannot be recorded in the real world. Here, we demonstrate that NMP can also be applied to real-world simulation by utilizing the transferability of neural networks.

Here we consider a new scenario, where a cube drops and bounces under an unknown external force and unknown material (softer than the CUBE dataset) and only position is accessible. Since such a partially observable simulation cannot provide intermediate physics for two-stage training, we adopt the pretrain-finetune paradigm of deep learning and pretrain NMP on the simulated CUBE dataset for a well-modularized model and then finetune it solely based on the position information. Figure 6 shows that NMP can still generate simulations accurately. This result not only justifies the practicality of NMP but also highlights the advantage of neural simulators in transferability.

### 4.3 UNIQUE BENEFITS OF MODULARIZATION

Notably, it is not easy to guarantee that a complex neural modular network works as we expected. Without the elaborative physical-aligned architecture and specialized training strategy in NMP, we cannot realize the unique benefits of modularization. To provide an intuitive understanding, we compare our two-stage approach against joint training of both networks from scratch in

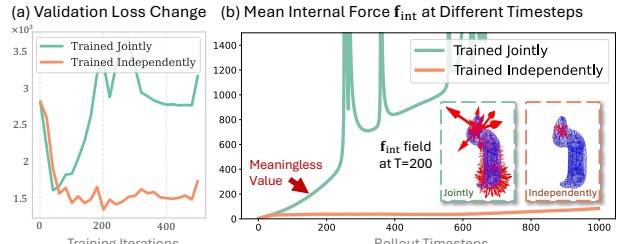

Figure 7: Comparison between joint training two modular networks and our two-stage independent training strategy in BOB.

Fig. 7. It can be observed that training NMP with two modular networks jointly will make the interface quantity $\mathbf{f}_{\text{int}}$ physically meaningless, which presents a noisy force field with impossibly large values. This confused intermediate representation indicates the modular network does not work as we expected, namely, the collapse issue (Mittal et al., 2022). In contrast, our special training strategy can help the modular network learn realistic physical quantities and stabilize the training process. As a result, the properly specialized modular network empowers the model with interesting features.

**Direct physical constraint** As stated before, our modularization maintains strict physical alignment with traditional simulators, enabling seamless physical constraint (Eq. 4). This feature cannot be accomplished in previous monolithic simulators. To further present the benefit brought by

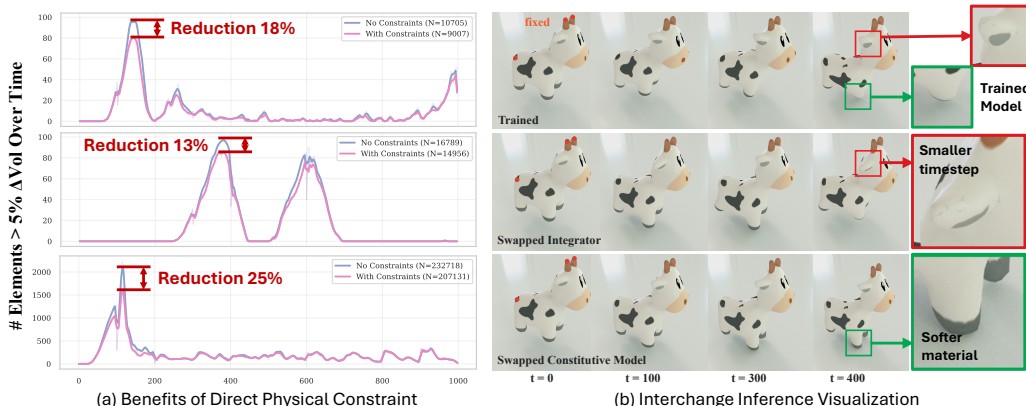

Figure 8: **(a)** *Impact of physical constraints* on three trajectories in CUBE. The curve records the number of tetrahedra exceeding 5% volume change at each time step and the legend counts the total number violating during 1,000 steps. **(b)** *Interchange inference.* Top: Sequence generated by our trained base model. Middle: Replacing the neural integration module with a semi-implicit integrator with a much smaller timestep. Bottom: Replacing the neural constitutive module with a softer St. Venant material increases deformation, visible in the foot position.

incorporating physical constraints, we compare our full method against a variant trained without the volume preservation losses, where the number of tetrahedral elements that exceed 5% volume change during simulation is recorded. As presented in Fig. 8(a), the model trained with local volume preservation shows better physical soundness. Specifically, the loss can further reduce the maximum number of violations by 10%. Despite the simulation of NMP still being imperfect, the special modularization provides us with an interface to explicitly constrain physical rules.

**Interchange inference** Benefitting from our specialized design of modular interfaces, NMP can seamlessly achieve interchange inference with traditional simulators. To verify the interchange stability, we first swap the neural integrator for a semi-implicit numerical integrator with a tiny timestep ($5 \times 10^{-6}$), $100\times$ smaller than during training (Fig. 8(b), middle). This captures higher-frequency dynamics, such as Spot's ear motion, showing that our learned constitutive module remains compatible with alternative numerical integrators. Second, we replace the neural constitutive model with a softer St. Venant material (Young's modulus $5 \times 10^4$), producing greater deformation (Fig. 8(b), bottom). These experiments highlight our framework's plug-and-play flexibility across integrators and materials—enabling fast adaptation to new behaviors without retraining.

## 5 CONCLUSION AND DISCUSSION

This paper presents *Neural Modular Physics* (NMP), a thorough modular framework that unites the physical reliability of classical simulators with the scenario flexibility of neural simulators. Empowered by specialized architecture and a two-stage training strategy, NMP successfully transforms the whole computation flow of elastic simulation into successive neural modules. The modular physics design presented in this paper (i) exposes intermediate physical quantities, (ii) permits plug-and-play interchange between analytic and learned implementations, and (iii) enables analytic computation of physical constraints, which are all impossible in previous monolithic neural simulators. In experiments, NMP demonstrates better accuracy and long-horizon stability compared to both neural and hybrid approaches, while maintaining good generalization across different initial conditions and resolutions, as well as presenting better scenario flexibility and efficiency than traditional simulators.

This paper only focuses on elastic simulation, which is already a broad and fundamental domain, spanning applications in graphics, robotics, biomechanics, and materials science. In this future, we would like to further extend the modular idea in broader physics, such as fluid simulation and multi-object interaction. The traditional fluid simulation is also natively modularized and typically contains four steps: advection, force process, pressure projection and boundary process, which provides insights for the neural module design. As for multi-object interaction, which is rarely explored in previous papers, a neural or traditional collision module is necessary to handle complex contact.

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

## A SUPPLEMENTARY WEBSITE

An anonymous supplementary website is available at https://sites.google.com/view/neural-modular-physics. It includes videos demonstrating our method's performance across all test scenarios, visualizing predicted dynamics under various initial conditions and resolutions.

## B IMPLEMENTATION DETAILS

In this section, we will describe more details of our experiments in the main text.

### B.1 GROUNDTRUTH SIMULATOR

We implemented our ground truth simulator in PyTorch to enable end-to-end differentiability. The simulator employs a semi-implicit integration scheme:

$$\mathbf{v}_{t+1} = \mathbf{v}_t + dt(\mathbf{M}^{-1}\mathbf{f} + \mathbf{g}) \tag{5}$$

$$\mathbf{x}_{t+1} = \mathbf{x}_t + dt\mathbf{v}_{t+1}, \tag{6}$$

where $dt = 5 \times 10^{-4}$ seconds denotes the timestep, $\mathbf{M}^{-1}$ represents the inverse mass matrix, $\mathbf{g}$ is gravity acceleration that is set to $-9.8 m/s^2$ in our experiments.

After integration, the simulator handles external constraints. For scenes involving ground contact (e.g., CUBE scene), vertices that fall below the ground plane are projected back to ground level with their vertical velocities set to zero. For scenes with fixed boundary conditions (BOB and SPOT), designated vertices are maintained at their rest positions with zero velocity throughout the simulation.

### B.2 DATASET GENERATION

**Timestep setting** For the first four environments, ground truth trajectories are generated with a small internal timestep of $dt = 10^{-5}$ seconds (50 substeps per output frame) to ensure stability. Each trajectory contains 1000 frames at a coarser resolution of $dt = 5 \times 10^{-4}$, which corresponds to one output frame per 50 simulation steps. Importantly, our model is only trained and evaluated on these coarser frames, meaning the neural integrator operates at the larger effective timestep of $dt = 0.0005$. For the FRICTION environment, we generate and test the simulator at $dt = 0.0001$ to test a larger timestep.

The above setup inherently challenges the model to predict dynamics over long time intervals, encouraging temporal generalization. While training uses rollouts of up to 200 steps, all reported results in the main text, including the 1000-step predictions, require the model to generalize far beyond its training horizon.

**Material configuration** For dataset generation of the first four environments, we used the neo-Hookean material model:

$$W_{NH}(\mathbf{F}) = \frac{\mu}{2}(I_C - 3) - \mu \ln(J) + \frac{\lambda}{2}(\ln(J))^2, \tag{7}$$

where $\mu$ and $\lambda$ are Lamé parameters, $J = \det(\mathbf{F})$ is the volume change ratio, and $I_C = \text{tr}(\mathbf{F}^T\mathbf{F})$ is the first invariant of the right Cauchy-Green deformation tensor.

Scene-specific material parameters were configured as follows. We configured different material stiffness across scenes while maintaining the same Poisson's ratio $\nu = 0.45$. The cube scene uses a stiffer material with Young's modulus $E = 5 \times 10^5$ Pa, while both Bob and Spot scenes use a more compliant material with $E = 1 \times 10^5$ Pa to allow for larger deformations.

For the FRICTION environment, we used the differentiable simulator DiffPD (Du et al., 2021), which is based on projective dynamics and thus requires a projective dynamics formulation of material energies. We refer the reader to the original paper for full details of the energy discretization and solver. For consistency with the other environments, we set the material parameters in the same manner as above, maintaining Poisson's ratio $\nu = 0.45$ while varying the Young's modulus $E$ to control material stiffness.

### B.3 Network architecture and training details

**Model architecture** The neural constitutive module is implemented as a two-layer MLP with 80 and 96 neurons, respectively. Our neural integration module predicts per-vertex velocity updates $\Delta \mathbf{v}$ given positions $\mathbf{x}$, velocities $\mathbf{v}$, and internal forces $\mathbf{f}_{\text{int}}$. Each of the three inputs is processed independently by a one-layer MLP with SiLU activation, producing embeddings of 32 dimensions respectively. Additionally, we use a learned alignment module (TNET) to produce aligned input mesh coordinates: it outputs a $3\times3$ rotation matrix (via a quaternion regressor) that aligns each mesh to a canonical frame. This promotes pose invariance and improves generalization to varied initial conditions. The aligned mesh positions, original mesh positions, velocities and forces are concatenated with the original inputs and passed through a three-layer MLP with 64 hidden units to regress $\Delta \mathbf{v}$. To regularize early training, we clamp predicted velocities to a maximum of 30 m/s in magnitude, following MeshGraphNet (Pfaff et al., 2020).

**Learned alignment module** To promote generalization across varying initial poses, we use a learned spatial alignment module that normalizes input positions by predicting a global $3\times3$ linear transformation matrix per sample. The input point cloud $\mathbf{x} \in \mathbb{R}^{N\times3}$ is first encoded by a shared MLPNET applied independently to each vertex. The resulting per-point features are aggregated via max pooling, then passed through a two-layer MLP to regress a 9-dimensional vector, reshaped into a $3\times3$ matrix. To encourage stability, the predicted matrix is initialized near the identity. Unlike prior approaches that constrain the output to be a rotation (e.g., via quaternions or projection to SO(3)), we allow a full linear transformation, enabling the network to learn scale, shear, and other canonicalizing deformations directly from data.

**Training configurations** Both networks are trained using the Adam optimizer with a cosine annealing learning rate schedule. The training process consists of two phases: independent pretraining of each network for 100 epochs, followed by joint fine-tuning for another 100 epochs. During pretraining, each network is trained with ground truth inputs from the simulation dataset, while fine-tuning allows the networks to adapt to each other's learned behaviors.

To stabilize sequence training and prevent error accumulation, we employ a teacher forcing strategy during both pretraining and fine-tuning. When processing sequences within each epoch, the simulation state is periodically reset to ground truth values. The reset interval increases from 60 to 200 steps following a cosine annealing schedule, gradually encouraging the networks to handle longer sequences of predictions without intervention.

### B.4 Baseline Implementation

We implement three representative neural simulators, Transolver (Wu et al., 2024b), EGNO (Xu et al., 2024), and MeshGraphNet (MGN) (Pfaff et al., 2020). To ensure fairness and reproducibility, all models are built on *their official codebases or libraries* and follow the recommended preprocessing and data pipelines where applicable. All baselines are optimized with the normalized mean squared error (NMSE) loss $\text{NMSE}(\hat{\mathbf{y}}, \mathbf{y}) = \frac{\|\hat{\mathbf{y}} - \mathbf{y}\|_2^2}{\|\mathbf{y}\|_2^2}$, using a fixed learning rate of 0.1 and training for 100 epochs. The following are detailed configurations:

- **Transolver** (Wu et al., 2024b): we adopt the Transolver Irregular Mesh variant and use 15 Transolver blocks with model width 256, 4 attention heads, an MLP ratio of 2, and dropout 0.1, while setting the slice number to 32 and the number of reference tokens to 16.

- **EGNO** (Xu et al., 2024): 15 layers with hidden size 256, 4 vector-message heads, dropout 0.1, and an RBF embedding of dimension 12.

- **MeshGraphNet** (Pfaff et al., 2020): 15 message-passing steps with hidden size 128.

To improve robustness and stabilize normalization statistics, we inject small Gaussian noise into the inputs of *all* three neural simulators (Transolver, EGNO, and MeshGraphNet) during training. Concretely, at each step, we perturb positions and velocities as $x_t \leftarrow x_t + \varepsilon_x$, $v_t \leftarrow v_t + \varepsilon_v$ with $\varepsilon_x, \varepsilon_v \sim \mathcal{N}(0, \sigma^2)$, where we use $\sigma = 3 \times 10^{-3}$.

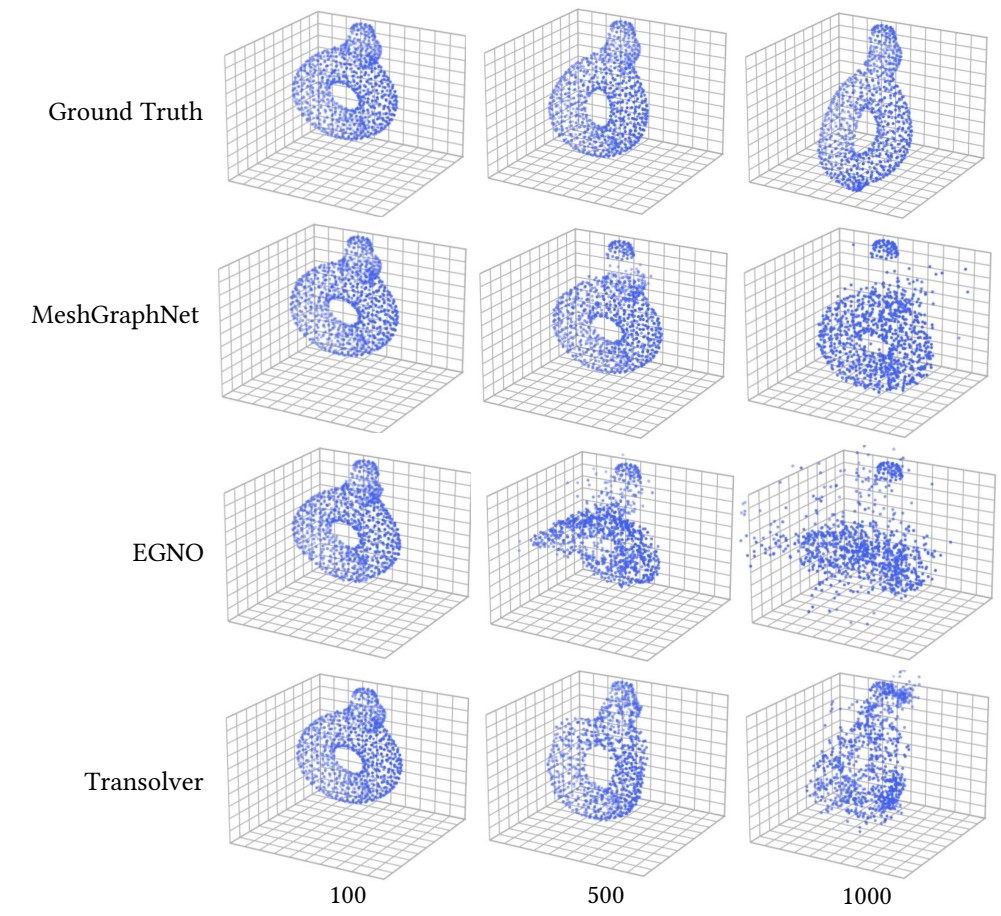

Figure 9: Qualitative comparison of rollouts for the finite element ground truth and three baselines at 100, 500, and 1000 steps on Bob. From top to bottom, rows show Ground Truth, MeshGraphNet, EGNO, and Transolver. Columns show snapshot frames at 100, 500, and 1000 steps.

For the hybrid physics simulator, we directly follow their official configurations for the model architecture configuration of NCLaw (Ma et al., 2023). For the physics simulator baseline, we use our groundtruth simulator described in Sec. B.1.

## B.5 Interchange Inference Experiment Settings

For material model swapping experiments, we used the St. Venant-Kirchhoff model:

$$W_{StVK}(\mathbf{F}) = \frac{\lambda}{2} \left(\text{tr}(\mathbf{E})\right)^2 + \mu\text{tr}(\mathbf{E}^2), \tag{8}$$

where $\mathbf{E} = \frac{1}{2}(\mathbf{F}^T\mathbf{F} - \mathbf{I})$ is the Green strain tensor.

## C Supplementary Results

In this section, we will provide more visualizations and results as a supplement to the main text.

### C.1 Comparison with Baselines

**Visual comparison**   As presented in Fig. 9, we compare Transolver (Wu et al., 2024b), EGNO (Xu et al., 2024), and MeshGraphNet (MGN) (Pfaff et al., 2020) against the ground truth at 100, 500, and 1000 steps. The visualizations reveal several distinct patterns in the baselines' performance.

Among the three neural simulators, Transolver best preserves geometry at short horizons. At 100 steps, its simulated shape and pose remain close to the reference, and by 500 steps, the results still appear coherent, following the trajectory with only moderate drift. However, by 1000 steps, Transolver loses local structure, leaving only a rough silhouette that reflects cumulative error and the loss of elastic memory.

EGNO deteriorates much earlier. By 500 steps, the configuration has already sagged into a blurred particle cloud with a strong downward bias, and by 1000 steps, the object becomes largely amorphous, with the original geometry no longer discernible.

MeshGraphNet maintains a recognizable outline longer than EGNO, but its motion is strongly over-damped. Oscillations quickly fade, contact responses smear, and the particles exhibit a spurious sinking or condensing trend rather than elastic rebound. By 1000 steps, the geometry stretches along the vertical direction and global coherence breaks down.

These results demonstrate the difficulty for monolithic neural simulators to accurately capture long-horizon physical dynamics. In contrast, as shown in Fig. 3, our proposed neural modular physics framework is able to generate stable and physically consistent simulations over extended horizons.

**New autoregressive prediction** In neural simulator baselines, their official configuration is to adopt the one-step autoregressive prediction. Thus, we also follow this setting in our experiments for both our method and baselines. One possible way to enhance these baselines is to make the model predict 100 steps at once; then the 1,000-step simulation only requires 10 times of rollout prediction, which is a practical trick to reduce the accumulation error.

Table 3: Compare with enhanced baselines, where each model will directly predict 100 steps at one inference. The RMSE of 1,000 steps is recorded.

| Method (BOB simulation) | RMSE |
| --- | --- |
| Transolver (2024b) (predict 100-steps) | 0.75 |
| EGNO (2024) (predict 100-steps) | 2.00 |
| MeshGraphNet (2020) (predict 100-steps) | 2.61 |
| **NMP (Ours, predict one-step)** | **0.58** |

Here, we also compare with enhanced baselines. As listed in Table 3, directly predicting 100 steps can reduce the long-term simulation error of neural simulators, especially Transolver, whose RMSE is reduced from 1.782 (Table 1) to 0.75. However, these baselines still fall behind our method, which highlights the advancement of modularization.

## C.2 INITIAL CONDITION DISTRIBUTION ILLUSTRATION

As stated in the main text, different simulation trajectories begin with distinct initial conditions, which naturally lead to divergent dynamic trajectories. To provide an intuitive illustration of this effect, we plot 40 trajectories from the CUBE dataset, including 32 cases from the training set and 8 cases from the test set. In these cases, variations arise from different initial velocities and cube orientations. As shown in Fig. 10, even small differences in initial velocity or orientation at the first step can significantly alter subsequent dynamics, creating a pronounced gap between training and test trajectories. This visualization highlights the inherent challenge of generalization in our experiments.

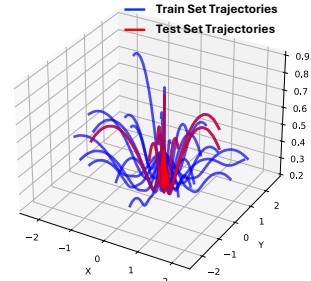

Figure 10: Visualization of trajectories with different initial conditions on CUBE.

## C.3 GENERALIZATION TO UNSEEN INITIAL CONDITIONS

Fig. 11 shows our method's generalization capability on the Spot scene across different initial conditions. Similar to the Bob scene discussed in the main text, we evaluated the model on multiple unseen initial velocities. The results demonstrate consistent performance across different scenarios - each row shows a unique trajectory where the model maintains stable motion while accurately preserving the fixed-point boundary conditions at both head and tail. At $t = 1000$, our predictions (bounded by blue boxes) closely match the ground truth configurations (within green boxes), indicating robust generalization despite the challenging dual-fixed-point constraints.

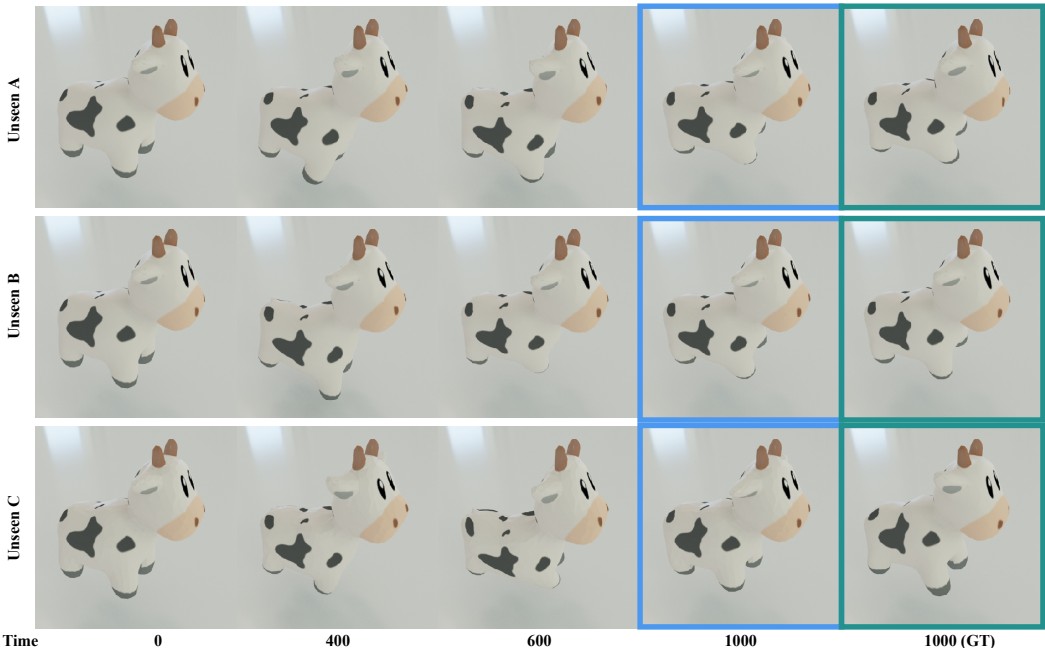

Figure 11: Generalization to unseen initial conditions for SPOT. Each row shows a sequence with different initial velocities, demonstrating our model's ability to handle varied dynamics while maintaining fixed-point constraints. Our predictions closely match the ground truth at t=1000.

## C.4 STANDARD DEVIATIONS FOR TABLE 1

Table 4 reports the per-sequence standard deviations (over 8 validation trajectories) corresponding to the performance means listed in Table 1.

Table 4: Standard deviations of RMSE for Table 1 (computed over 8 validation trajectories).

| Method | CUBE | | | CUBE XL | | | SPOT | | | BOB | | |
|---|---|---|---|---|---|---|---|---|---|---|---|---|
| | 100 | 500 | 1000 | 100 | 500 | 1000 | 100 | 500 | 1000 | 100 | 500 | 1000 |
| Transolver | ±0.069 | ±0.364 | ±0.706 | ±0.073 | ±0.726 | ±2.323 | ±0.062 | ±0.343 | ±0.673 | ±0.043 | ±0.230 | ±0.563 |
| EGNO | ±0.197 | ±0.979 | ±2.001 | ±0.103 | ±0.815 | ±1.533 | ±0.098 | ±0.333 | ±0.273 | ±0.110 | ±0.524 | ±1.038 |
| MeshGraphNet | ±0.002 | ±0.009 | ±0.002 | ±0.010 | ±0.034 | ±0.631 | ±0.001 | ±0.003 | ±0.001 | ±0.001 | ±0.001 | ±0.002 |
| **Ours** | ±0.010 | ±0.046 | ±0.161 | ±0.010 | ±0.149 | ±0.107 | ±0.001 | ±0.135 | ±0.162 | ±0.021 | ±0.136 | ±0.348 |

## C.5 PHYSICS VALIDATION METRICS

In addition to RMSE, the physical soundness of simulators is also an essential metric. Therefore, as a supplement to Figure 8, we also count the number of tetrahedra exceeding 5% volume change during a 1,000-step rollout, which measures the local volume preservation of different simulators. As shown in Table 6, NMP surpasses all the other baselines in this physics validation metric. These results further demonstrate the benefits of neural modular physics in guaranteeing physical soundness.

Table 5: Physics validation of different models, where we record the number of tetrahedra exceeding 5% volume change. A smaller number indicates better physical soundness.

| Method (CUBE simulation) | Number |
|---|---|
| Transolver (2024b) | 1271827 |
| EGNO (2024) | 1595317 |
| MeshGraphNet (2020) | 479566 |
| **NMP (Ours)** | **231094** |

## D ABLATION STUDY

Here, we include some ablations about the hyperparameters and configurations of neural modules.

**Model complexity analysis**    We ablate the size of the neural constitutive model to assess sensitivity to network complexity. Our original architecture (2 hidden layers, 96 and 80 neurons) was chosen to maximize GPU memory usage during training. Reducing the architecture to 2 layers of 64 and 40 neurons, and evaluating on the Cube dataset with a ground-truth integrator, we observe a higher 1000-step RMSE ($0.337\pm0.270$) compared to the original model ($0.220\pm0.174$), highlighting the importance of model size. One promising direction is to further scale up the NMP model.

**Hyperparameter analysis**    As stated in Eq. 4, adding physical constraint also introduces a weight hyperparameter. In the experiments, we selected the volume regularization constant such that the constraint term contributed approximately 1% to the total loss during training. This choice reflects a balance between encouraging physical plausibility and preserving learnability. Here, we evaluated model performance under a range of volume loss weights $\{0, 1, 10, 1000\}$. As shown right, moderate regular-

Table 6: Physical constraint loss weight analysis. The following test is conducted on the "real-world" CUBE scenario in Figure 6.

| Loss Weight | 1000-step RMSE |
| --- | --- |
| 0 | 0.397 |
| 1 | 0.337 |
| 10 | 0.604 |
| 1000 | 2.579 |

ization (e.g., 1.0) improves long-horizon accuracy, while excessively large values (e.g., 1000) degrade performance because large weight would discourage mesh achieving desired deformation.

## E    FAILURE MODE ANALYSIS

To further demonstrate the simulation property of our model, we select the worst case (with the largest RMSE) among the BOB task. The visual comparison is included in Figure 12. It is observed that even in the worst case, NMP still accurately simulates the main dynamics of Bob. Besides, we can also notice that dynamics with large deformation are really challenging for simulation.

## F    LIMITATIONS

In this paper, we have proposed a new framework to modularize elastic simulation into well-designed and well-optimized neural networks, which brings several unique advantages in scenario flexibility and physical constraint. Although neural modular physics is supposed to be a general idea, our current experiments only cover the elastic simulation, as stated in this paper's title. Adapting the framework to other physical domains, such as fluid dynamics, will be a promising direction. However, fluid simulation will need a distinct modularization architecture to fit the pipeline of traditional computational fluid dynamics. Thus, we pinpoint our current scope only in elastic simulation and would like to leave the exploration of NMP in other domains as future work.

LLM Usage Declaration: LLMs were used solely to assist in writing and language polishing.

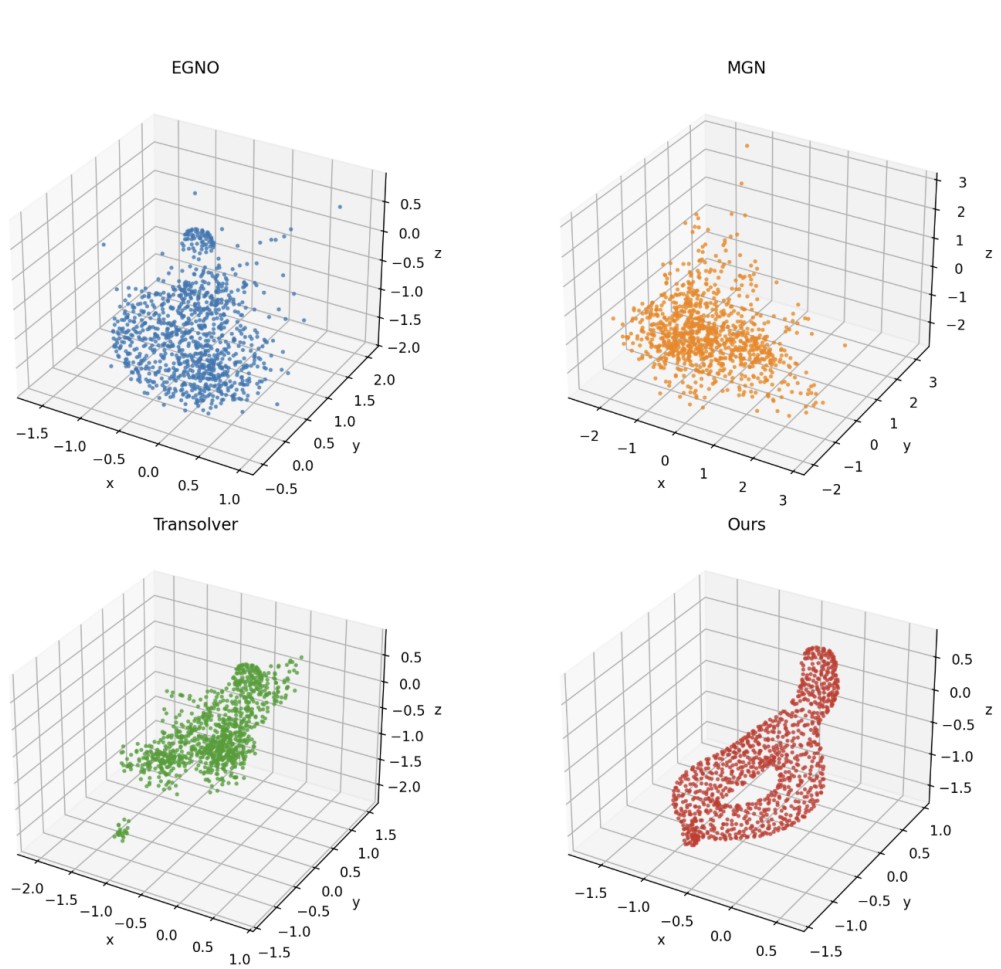

Figure 12: Failure mode analysis. We visualize the last frame in the worst case, which is selected according to the RMSE of NMP. The predictions of other baselines are also included.

