# OpenReview forum: "Neural Modular Physics for Elastic Simulation"
_ICLR.cc/2026/Conference — Submitted to ICLR 2026_

### Official Review · Reviewer_b4yZ · 2025-10-29

**Soundness:** 3
**Presentation:** 3
**Contribution:** 3
**Rating:** 8
**Confidence:** 5

**Summary:**

The paper Neural Modular Physics for Elastic Simulation , introduces a fully modular neural simulator for elastic dynamics. This represents a new direction relative to the monolithic, end-to-end neural simulation paradigms (NO, GNN) and hybrid simulators that replace partial components demonstrated through a complete modular neural architecture that mirrors traditional finite element method (FEM) computation flows. The approach enables direct supervision of intermediate physical quantities and interchangeable traditional-numerical and neural components, which is a novel result in the neural simulation literature. The contribution of this paper lies more in systematic integration and training methodology (the two-stage modular physics training) than in the individual components.
Given the surge of interest in physics-informed ML, differentiable simulation, and scientific machine learning. The paper addresses a key pain point of neural simulators—lack of interpretability and physical soundness—making it relevant to computational physics and graphics communities as well. The papers focus on elastic simulation is narrower than universal PDE solvers, but serves as an excellent proof-of-concept domain.

**Strengths:**

1. Concept:
The paper provides a fairly new approach to learning-based, physically interpretable simulation approach that modularizes elastic dynamics into neural subcomponents aligned with traditional numerical solvers. The idea of decomposing physics simulation into learnable yet interpretable modules, supervised via intermediate physical quantities, represents a meaningful conceptual and architectural advance.
While the idea of combining neural networks with modular solvers has appeared before the innovation lies more in systematic integration and training methodology (the two-stage modular physics training) than in the individual components.
2. Practicality/Application:
When combining neural networks with physics-based modular solvers, one of the hardest problems is ensuring boundary condition consistency between modules that are partly data-driven and partly physically constrained. In hybrid or modular networks, each subnetwork (e.g., constitutive law, integrator) may implicitly assume slightly different boundary behavior, leading to inconsistencies at their interfaces — this is known as a boundary-condition coupling error.  The NMP framework proposes to address this through architectural and Training level strategies.  The Neural Integration Module preserves the same interface as an implicit FEM integrator but replaces the internal update terms with a neural network. After the neural update, boundary condition enforcement and collision handling are explicitly re-applied.
Hence, NMP preserves:
Dirichlet BCs
Neumann BCs
Contact constraints
This design ensures neural predictions remain compatible with traditional BC enforcement — solving the “boundary leakage” problem found in prior hybrids
3. Results: Published results show improved performance over top baselines as well as improved stability (reduced collapse)
4. Generalizability: is demonstrated against unseen initial conditions and higher mesh resolutions.
5. Evidence: The SPOT and BOB experiments are specifically designed to test BC handling
SPOT: multiple fixed boundary points (head and tail)
BOB: fixed vertices (head) and free elastic regions

**Weaknesses:**

1. Practicality/Application:  solution process and validation
Computational cost and scalability are not discussed (e.g., time vs. FEM or other neural simulators). This is a critical metric to be evaluated.
Lack of error analysis: No discussion on failure modes or interpretability visualization for intermediate variables.  This is also critical for real world applications on elastic body problems.
2. Results: are limited to non real world elastici body  FEM problems
3. Generalizability: is limited to elastic solids only. Extensions to fluids or multi-physics are not shown and this gap is acknowledged by the authors.
4. Evidence and support: the following weaknesses exist:
No ablations on module architecture: How sensitive is performance to neural constitutive/integration model complexity?
Limited physics validation metrics: Energy conservation, stress-strain correlation, or physical invariants could further substantiate realism.

Writing clarity:
line 121 -"dynamics on top of analytical dynamics to account for unmodeled" ... is an incomplete sentence
line 135: "In this paper, we notice the internal modularity". ... please fix
line 141: "As aforementioned," --> as mentioned earlier ?
line 160 : "derives two disentangled modules" --> two decoupled ?
line 182: "with vertice position" --> vertex position ?
line 329: Transolver (?). --> what is the "? " provide citation, Wu ?

**Questions:**

Please address the following practical issues
1. Discuss time to converge relative to traditional solvers on a real world engineering problem if possible
2. discuss error analysis and boundary condition matching
 3. provide more evidence on physics validation metrics: Energy conservation, stress-strain correlation, or physical invariants could further substantiate realism.

---

> ### Comment · Reviewer_b4yZ · 2025-11-25
>
> i do not see any comments by the authors to the questions raised by myself and the other reviewers. Earlier I had given a rating of "Accept" based on the assumption that the authors would provide responses to questions asked by me regarding improvements to the paper.  in my opinion these were easy fixes and would add value to the paper and for the reader.
> Given that no attempt was made on any responses and question remain unresolved, I am dropping my score to 6.

---

> > ### Comment · Reviewer_b4yZ · 2025-11-26
> >
> > i revised my scores to original, author reviews came in and are generally satisfactory and address all concerns

---

> > > ### Author Response · Authors · 2025-11-27
> > > **Thanks for your response and support for our work**
> > >
> > > Dear Reviewer b4yZ,
> > >
> > > We sincerely thank you for your time and effort in reviewing our paper. Your insightful feedback and constructive suggestions are greatly appreciated and have been invaluable in guiding our revisions.
> > >
> > > We are truly grateful for your support.

---

> ### Author Response · Authors · 2025-11-26
>
> Dear Reviewer b4yZ, we are currently incorporating several additional experiments and addressing all the raised questions in detail with added experiments in the forms of diagrams and tables. The full responses with updated paper will be finalized and submitted very soon. We apologize for the delay and thank you in advance for your patience.

---

> ### Author Response · Authors · 2025-11-26
> **Response to Reviewer b4yZ (Part 1)**
>
> Many thanks to Reviewer b4yZ for acknowledging our contribution and providing a detailed review and insightful suggestions. All the rebuttal has been included in the $\underline{\text{revised paper}}$.
>
> > **W1:** "Practicality/Application: solution process and validation Computational cost and scalability are not discussed (e.g., time vs. FEM or other neural simulators). This is a critical metric to be evaluated. Lack of error analysis: No discussion on failure modes or interpretability visualization for intermediate variables. This is also critical for real world applications on elastic body problems."
>
> Following your suggestion, we have included all the requested experiments during the rebuttal.
>
> **(1) Efficiency comparison with baselines.**
>
> As shown below, our method, a thorough modular model, is significantly faster than pure physics or neural-physics hybrid methods. And NMP's efficiency is also comparable to other neural baselines.
>
> | Bob Simulation Task    | Inference time 1000 steps (ms) |
> | - | - |
> | Transolver (2024)     | 11796                          |
> | EGNO (2024)           | 26351                          |
> | MeshGraphNet (2020)   | 27603                          |
> | NCLaw (2023)          | 97121                               |
> | Traditional Simulator | 90275                          |
> | NMP (Ours)            | 2250                           |
>
> **(2) Interpretability visualization for intermediate variables.**
>
> We have included the visualization of internal force in $\underline{\text{Figure 6(b) of original submission}}$.
>
> **(3) Discussion on failure modes**
>
> We have selected the worst case from the test set according to the RMSE of NMP as the failure modes. As shown in $\underline{\text{Figure 12 of Appendix E revised paper}}$, the failure modes correpond to large deformations, making the prediction more challenging. It is worth noticing that, in this case, NMP still outperforms other baselines.
>
> > **W2:** "Results: are limited to non real world elastici body FEM problems"
>
> Thanks for your detailed review.
>
> **(1) NMP can be extended to real-world simulations.**
>
> Here, we include an additional experiment  to illustrate a "Simulation to Real" pipeline by introducing unknown material and force variations in a controlled environment:
> | Experiment setting  | Accessible physics                     | Underlying dynamics                                          |
> | - | - | - |
> | Simulation scenario | Internal forces, position and velocity | Cube scenario |
> | Real scenario       | Position                  | Cube scenario with unknown softer material and external force |
>
> It is easy to adopt a pretrain-finetuning pipeline for the above sim2real task:
>
> - During the pretraining stage, we optimize the model based on the Cube dataset. Then, train a neural modular physics network based on internal force, vertex position and velocity, which is sufficient to deliver a successful modularized NMP model.
> - During the finetuning stage, we directly fine-tune the entire neural modular physics network only based on vertex position, which can be obtained in the real world.
>
> In this way, NMP can be applied to real data and maintain a good modularization. The visual comparison is provided in $\underline{\text{Figure 6 of revised paper}}$.
>
> **(1) About limitation in FEM.**
>
> In this paper, we mainly focus on the elastic simulation, where the FEM is the mainstream simulation method. As for other methods, such as FVM, we have proposed a possible design in the next question.

---

> ### Author Response · Authors · 2025-11-26
> **Response to Reviewer b4yZ (Part 2)**
>
> > **W4:** "Evidence and support: the following weaknesses exist: No ablations on module architecture: How sensitive is performance to neural constitutive/integration model complexity? Limited physics validation metrics: Energy conservation, stress-strain correlation, or physical invariants could further substantiate realism."
>
> Thanks for your valuable suggestion. We have included all the requested experiments.
>
> **(1) Ablations on model complexity.**
>
> We ablate the size of the neural constitutive model to assess sensitivity to network complexity. Our original architecture (2 hidden layers, 96 and 80 neurons) was chosen to maximize GPU memory usage during training. Reducing the architecture to 2 layers of 64 and 40 neurons, and evaluating on the Cube dataset with a ground-truth integrator, we observe a higher 1000-step RMSE (0.337 ± 0.270) compared to the original model (0.220 ± 0.174).
>
> The above experiments highlights the importance of model size in neural simulators. One promising future direction is to further scale up the NMP model.
>
>
> **(2) Adding physics validation metrics.**
>
> We have included a new metric to measure the Local Volume Preservation, which counts the number of elements with over 5% ΔVol Over Time.
>
> | Bob Simulation | >5% ΔVol Over Time |
> | - | - |
> | Transolver                          | 1271827       |
> | EGNO                          |   1595317     |
> | MeshGraphNet                          |  479566      |
> | NMP (Ours)                          |    231094    |
>
> > **W5:** All the writting issues.
>
> Thanks for your detailed review. We have resolved all these typos in the $\underline{\text{revised paper}}$.
>
> > **Q1:** "Please address the following practical issues" (1) Discuss time to converge relative to traditional solvers on a real world engineering problem if possible, (2) discuss error analysis and boundary condition matching, (3) provide more evidence on physics validation metrics: Energy conservation, stress-strain correlation, or physical invariants could further substantiate realism.
>
> Thanks for your questions. Here are our clarifications.
>
> 1. Since NMP is a thorough modular network, it can generate the simulation results solely based on the forward pass of the neural network, which does not need an iterative process for simulation and also does not have a convergence issue.
>
> 2. We have included the error analysis in the revised paper. Check **W1** for details. Regarding the boundary condition matching, as visualized in $\underline{\text{Figure 6 of original submission}}$, it is really hard to learn a meaningful interface quantity between the neural constitutive and integration modules. Based on the two-stage training strategy, NMP can ensure a good match on the interface physics.
>
> 3. We have newly measured the Local Volume Preservation of different models. Please check **W4** for results.

---

### Official Review · Reviewer_ZLNW · 2025-10-31

**Soundness:** 3
**Presentation:** 3
**Contribution:** 2
**Rating:** 6
**Confidence:** 3

**Summary:**

The paper proposes a hybrid-physics architecture and training method that achieves better physical consistency and generalizability than traditional simulators and neural models for elastic physics rollout. The modular design for elastic physics is inspired by the FEM method.

**Strengths:**

- The experiments provide good empirical evidence that their NMF model has better long term rollout performance for both seen and unseen scenarios in comparison to the baselines.
- Their NMF model allows the addition of soft physics constraints in a fairly straightforward way.
- The paper provides a thorough explanation of their modular model design and reasoning.

**Weaknesses:**

- The architecture is specialized for elastic physics, which will not generalize to other equations and setttings. However, I acknowledge the idea of replacing components of a simulator with neural modules is an interesting idea.
- The paper proposes a way to replace parts of an FEM method with neural networks, but this is not straightforward to apply to other types of solver schemes such as finite difference methods and spectral methods.
- The strain-displacement matrix is required a priori to use the NMF model.

**Questions:**

- Do you plan to release the code for your model training and simulations?
- What are the costs in terms of flops and time of the NMF model inference vs the simulator used to generate ground truth trajectories?
- Since the neural constitutive model requires the $B_e$ matrix and $V_e$ values to work, do you provide these in your test scenarios to the model? Is the $B_e$ term easy to derive in some way or should it also be learned?
- How do you determine when to stop the separate training phase for each module? Do you train for a fixed number of steps or use some other condition for stopping?
- You use a regularization constant of 0.1 for the volume loss term you add during joint training. How did you select this value? What happens if you increase it to make the physical constraint stronger? Does the overall trajectory become more physically plausible or does your model fail to learn due to the difficulty of satisfying this constraint?
- There are few typos I noticed in the paper that can be corrected:
  - Line 121: "Examples include learned data-driven discretization stencils (Bar-Sinai et al., 2019) and learned residual
dynamics on top of analytical dynamics to account for unmodeled (Yin et al., 2021)." Maybe you want to write unmodeled dynamics?
  - Line 211: "Still start from classical FEM." Rewrite this to be a more clear introductory sentence.
  - Line 424: "With out the elaborative physical-aligned architecture and specialized training strategy in NMP, we cannot release the unique benefits of modularization." I think you mean to write "we cannot realize" instead of "we cannot release".

---

> ### Author Response · Authors · 2025-11-26
> **Response to Reviewer ZLNW (Part 1)**
>
> Many thanks to Reviewer ZLNW for providing the insightful review and questions. All the rebuttal has been included in the $\underline{\text{revised paper}}$.
>
> > **W1:** "The architecture is specialized for elastic physics, which will not generalize to other equations and setttings. However, I acknowledge the idea of replacing components of a simulator with neural modules is an interesting idea."
>
> We would like to thank the reviewer for acknowledging our idea.
>
> First, we respectfully clarify that elastic simulation itself is a broad and fundamental domain, spanning applications in graphics, robotics, biomechanics, and materials science. By demonstrating generalization across multiple unseen conditions, geometries, and mesh resolutions, our method already covers a wide spectrum of elastic simulation tasks.
>
> Moreover, **even traditional simulators separate methods by domain: elastic simulation often uses the Finite Element Method (FEM), while fluid simulation typically relies on the Finite Volume Method (FVM) or grid-based schemes such as semi-Lagrangian advection and pressure projection**. As noted in $\underline{\text{Appendix D of original submission}}$, we acknowledge this domain specificity and view the extension of NMP to other physical domains as exciting future work.
>
> Still, we emphasize that NMP is a framework, not a simulator for elastic-only phenomena. Its modular design enables generalization to other domains by swapping in domain-appropriate physics modules and neural components. Below, we outline a feasible NMP extension for fluid simulation, drawing from classical simulation methods such as Stable Fluids [Stam 1999].
>
>
> | NMP extension to Stable Fluids [Stam 1999] | Design | Role
> | - | - | -
> | Module 1               |     Neural Advection Module   | Approximates advection of velocity fields or scalars; replaces semi-Lagrangian advection with a learned transport map.
> | Module 2               |  External Forcing Module      |  Applies learned body forces (e.g., buoyancy or control inputs), possibly conditioned on
> | Module 3               |    Neural Pressure Projection    | Solves for divergence-free condition (∇·u = 0); replaces Poisson solver with a learned correction operator enforcing incompressibility.
> | Module 4               |  Boundary Condition Module      |  Impose solid/wall and free‑surface boundaries.
>
> > **W2:** "The paper proposes a way to replace parts of an FEM method with neural networks, but this is not straightforward to apply to other types of solver schemes such as finite difference methods and spectral methods."
>
> Thanks for pointing this out. In this paper, we mainly focus on the elastic simulation, where the FEM is the mainstream simulation method. As for other methods, such as FVM, we have proposed a possible design in **W1.**, which is also included in $\underline{\text{"Conclusion and Discussion" section of revised paper}}$
>
> > **W3:** "The strain-displacement matrix is required a priori to use the NMF model."
> >
> > **Q3:** "Since the neural constitutive model requires the B_e matrix and V_e values to work, do you provide these in your test scenarios to the model? Is the term easy to derive in some way or should it also be learned?"
>
> In our neural constitutive module, we follow the classical finite element method (FEM) approach to compute internal forces from stress via a strain-displacement matrix B_e, which encodes how local displacements of a finite element translate into strains and is central to computing internal forces in FEM-based simulations.
>
> Importantly, B_e depends **only on the undeformed mesh geometry and the analytical shape function gradients**, which are well-established for standard FEM elements (e.g., linear tetrahedra). Thus, B_e and the element volumes V_e can be **precomputed once from the reference mesh and reused at every simulation step**. In all our experiments, these terms are provided to the model exactly as in classical simulators — they do not need to be learned and are straightforward to obtain for any mesh.
> > **Q1:** "Do you plan to release the code for your model training and simulations?"
>
> We will release the code upon acceptance, including training data, model checkpoints, complete code and scripts.

---

> ### Author Response · Authors · 2025-11-26
> **Response to Reviewer ZLNW (Part 2)**
>
> > **Q2:** "What are the costs in terms of flops and time of the NMF model inference vs the simulator used to generate ground truth trajectories?"
>
> As per your request, we have included the efficiency comparison below, both implemented and tested on GPU. While NMP requires more FLOPs, its actual runtime is significantly faster than the traditional simulator. This is because the traditional FEM-based simulator employs implicit integration with up to 50 substeps per timestep to ensure numerical stability, resulting in high computational cost. In contrast, NMP is trained to make stable, single-step predictions at larger time intervals, avoiding the need for such fine temporal discretization. Furthermore, NMP's computation is entirely composed of parallelizable neural network inference, which benefits from highly optimized tensor operations on modern GPUs.
> | Bob Simulation Task   | Time for 1000 steps (ms) | Flops for 1000 steps (GFLOPs) |
> | - | - | - |
> | Traditional Simulator | 90275                    | 43                            |
> | NMP (Ours)            | 2250                     | 90                            |
>
>
>
>
> > **Q4:** "How do you determine when to stop the separate training phase for each module? Do you train for a fixed number of steps or use some other condition for stopping?"
>
> As stated in $\underline{\text{Section B.3 Training configurations}}$, we just separately train two modules for a fixed number, which is 100 epochs.
>
> > **Q5:** "You use a regularization constant of 0.1 for the volume loss term you add during joint training. How did you select this value? What happens if you increase it to make the physical constraint stronger? Does the overall trajectory become more physically plausible or does your model fail to learn due to the difficulty of satisfying this constraint?"
>
> We selected the volume regularization constant such that the constraint term contributed approximately 1% to the total loss during training. This choice reflects a balance between encouraging physical plausibility and preserving learnability.
>
> Following your suggestion, we evaluated model performance under a range of volume loss weights. As shown below, moderate regularization (e.g., 0.1, our current configuration) improves long-horizon accuracy, while excessively large values (e.g., 1000) degrade performance because large volume loss would discourage the mesh from achieving desired deformation:
>
> | Regularization constant | 1000-steps RMSE on Cube-RealWorld |
> | - | - |
> | 0.0    |   0.397  |
> | 0.1    |   0.337  |
> | 1     |   0.604  |
> | 100 |   2.579  |
>
>
> > **Q6:** "There are few typos I noticed in the paper that can be corrected:"
>
> Many thanks for your detailed review. All these typos have been resolved in the $\underline{\text{revised paper}}$.

---

### Official Review · Reviewer_BcK5 · 2025-11-03

**Soundness:** 3
**Presentation:** 3
**Contribution:** 3
**Rating:** 4
**Confidence:** 5

**Summary:**

The paper propose an neural simulator framework for elastic simulation. The framework follows closely the traditional FEM simulation pipeline, with two modules replaced by neural networks: per-element Piola–Kirchhoff stress tensors computation and per-vertex velocity increment computation.

**Strengths:**

- The overall presentation is clear.

**Weaknesses:**

- The framework seems just a replication of some groundtruth FEM simulation method. In separate trainings for each module, it seems that the groundtruth constitutive model law and the groundtruth time integration are both available. The two groundtruth pieces are basically all you need to implement the whole simulation in the traditional way. I am concerned about the motivation of the method: if you need to implement the whole traditional pipeline to get groundtruth, why bother replacing modules with neural networks? Are there any applications beyond replication?

- The framework seems not applicable on real data. The assumption of known per-vertex internal forces is not practical in reality.

- The framework seems to only work for interactions between single objects and a fixed environment.
    - The contact information, especially the offset of the ground plane, is stored in time integration network. Updated ground plane offset may not work.
    - And I don't think the neural time integration can work for multiple objects. The neural modeling of time integration is not aware of other objects.

**Questions:**

- How to make sure when F is a rigid transformation, the stress is zero? A unit test may need to make sure the simulation can maintain its rest shape forever if there is no external force.

---

> ### Author Response · Authors · 2025-11-26
> **Response to Reviewer BcK5 (Part 1)**
>
> We would like to sincerely thank Reviewer BcK5 for providing valuable feedback and questions. All the rebuttal has been included in the $\underline{\text{revised paper}}$.
>
> > **W1:** "The framework seems just a replication of some groundtruth FEM simulation method. In separate trainings for each module, it seems that the groundtruth constitutive model law and the groundtruth time integration are both available. The two groundtruth pieces are basically all you need to implement the whole simulation in the traditional way. I am concerned about the motivation of the method: if you need to implement the whole traditional pipeline to get groundtruth, why bother replacing modules with neural networks? Are there any applications beyond replication?"
>
> Thanks for your detailed comments and insightful question.
>
> First of all, we would like to highlight that **our motivation is NOT a replication of the traditional simulator but to integrate the advantages of both neural and physics simulators.** As stated in $\underline{\text{Lines 61-69 of Introduction}}$, "pure numerical methods are less adaptable in scenarios with unknown information". Thus, we attempt to "design neural simulators with a thorough modular architecture, maintaining both data-driven flexibility and physical soundness."
>
> With our specialized architecture and training strategy, NMP can support the following new applications over the traditional simulator:
>
> **(1) Scenario Flexibility: Simulation under unknown physics (e.g., environment frictional force).**
>
> In $\underline{\text{Figure 5 of original submission}}$, we have experimented with the unknown frictional force scenario. Although we performed system identification to obtain reasonable estimates of the physics simulator’s input parameters, both traditional and neural-physics hybrid methods have failed. This experiment clearly demonstrate the scenario flexibility of our method.
>
> **(2) Fast inference for long-term dynamics.**
>
> As shown below, NMP has significantly better inference efficiency, which is around 40x faster than the traditional simulator.
>
> | Bob Simulation Task   | Time for 1000 steps (ms) |
> | - | - |
> | Traditional Simulator | 90275                    |
> | NMP (Ours)            | 2250                     |
>
> > **W2:** "The framework seems not applicable on real data. The assumption of known per-vertex internal forces is not practical in reality."
>
> Thanks for your detailed review. As pointed out by the reviewer, our method requires internal force for separate training to avoid mode collapse.
>
> **However, NMP can be extended to real data.**  To address this concern directly, we have conducted a new Sim-to-Real proof-of-concept transfer experiment included in the revised paper (Fig. 6), demonstrating that NMP can successfully adapt to real-world scenarios where only partial physical information is available. We consider two settings:
>
> | Experiment setting  | Accessible Physics During Training                     | Ground-Truth Dynamics                                          |
> | - | - | - |
> | Simulation scenario | Internal forces, position and velocity | Standard cube dataset |
> | Real-world variation      | Position                  | Cube with softer material + unknown external force |
>
> This setup mimics a common real-world challenge: how to adapt a simulator to a physical system when only partial observations are available (e.g., 3D positions) and the internal dynamics are not fully known. It is easy to adopt a pretrain-finetuning pipeline for the above sim2real task:
>
> - During the pretraining stage, we optimize the model based on the Cube dataset. Then, train a neural modular physics network based on internal force, vertex position and velocity, which is sufficient to deliver a successful modularized NMP model.
> - During the finetuning stage, we directly fine-tune the entire neural modular physics network only based on vertex position, which can be obtained in the real world.
>
> In this way, NMP can be applied to real data and maintain a good modularization. The visual comparison is provided in $\underline{\text{Figure 6 of revised paper}}$.

---

> ### Author Response · Authors · 2025-11-26
> **Response to Reviewer BcK5 (Part 2)**
>
> > **W3:** "The framework seems to only work for interactions between single objects and a fixed environment." (1) The contact information, especially the offset of the ground plane, is stored in time integration network. Updated ground plane offset may not work. (2) And I don't think the neural time integration can work for multiple objects. The neural modeling of time integration is not aware of other objects.
>
> First of all, we want to highlight that multi-object interaction is indeed a challenging task. All of our neural baselines, including Transolver (2024), EGNO (2023) and MeshGraphNet (2020), do not experiment with this scenario or unfixed environment.
>
> One possible way to handle multi-object interaction is to integrate a traditional collision handler to NMP. Notably, as a modular architecture, NMP is of good flexibility and can be conveniently intergated or interchanged with traditional numerical modules, which has been verified in $\underline{\text{Figure 7 of original submission}}$. As this is a less explored topic in this direction, we would like to leave this as a future work.
>
> > **Q1:** "How to make sure when F is a rigid transformation, the stress is zero? A unit test may need to make sure the simulation can maintain its rest shape forever if there is no external force."
>
>  Thank you for this important question. Our neural constitutive model  $f_{\theta}$ is specifically constructed to ensure zero stress under rigid transformations, which is a key physical requirement. We achieve this invariance through the following architectural design choices:
>
> - Part 1: For the deformation gradient $\mathbf{F}_e$, subtract an identity matrix from it as the input data normalization， so that the zero-input condition corresponds to no deformation.
> - Part 2: Removing the bias term in all MLP layers of $f_{\theta}$, This guarantees zero output (i.e., zero stress) when  $\mathbf{F}_e=\mathbf{I}$, which corresponds to a rigid transformation with no strain.
> - Part 3: Using GELU activation function in $f_{\theta}$, which are smooth and centered around zero, further reinforcing the zero-stress condition at rest shape.
>
> Based on the above design, when $F$ represents a rigid transformation, namely $F_e$ is an identity matrix, the input of $f_{\theta}$ will be equal to zero after part 1 normalization. Further, since the neural constitutive module does not contain the bias term and adopts the GELU activation, the output of $f_{\theta}$ will also be zero, namely zero-stress.

---

### Official Review · Reviewer_7hFZ · 2025-11-03

**Soundness:** 3
**Presentation:** 3
**Contribution:** 2
**Rating:** 4
**Confidence:** 4

**Summary:**

This paper proposes to use data-driven methods to replace the traditional solvers in the elastic simulation. The experiments show the effectiveness of the proposed method, leading to low errors and stable long-horizon stability. The experiments also show the proposed method is better for unseen conditions or inputs. Another contribution is that since the proposed method supervises the intermediate variables, it means the proposed method can use physics knowledge as contraints.

**Strengths:**

1. This paper is easy to follow and well organized.
2. This paper evaluates their method from multiple aspects: 1. Comparison with neural operator based baselines. 2.  Joint training versus separate training. 3. Comparison with physics simulator. 4. Inference by combining with traditional simulators. 5. Comparison with no direct physical constraint included.

**Weaknesses:**

1. The biggest weakness is that the proposed method is just substituting the immediate two steps that are done by traditional methods with data-driven methods. Thus, novelty is the biggest issue to me. Besides, the neural constitutive step is following the existing paper.
2. Another contribution claimed by the authors is the separate training plus joint training. From what I understand, this training method was also proposed in the existing work to solve the “collapse” issue, which is also not new.
3. For the experiments, the authors compare it with purely data-driven methods. However, their method uses physics. I don’t think the experiments are good enough for showing the advantages of their methods. For example, they should compare with the traditional methods and we can see the gap when physics is used for both methods.
4. It will be good to see the advantages/disadvantages of the methods including baselines in terms of inference time.
5. The method is only tested on the specific problem, which means the scope is limited.

**Questions:**

1. For strengthening the experiments, do the authors think the baselines can augment with physics knowledge?
2. What is the error for predicting just one step forward for each method?
3. For baselines, have you tried non-autoregressive prediction, e.g., directly predicting all the following 100 steps.

---

> ### Author Response · Authors · 2025-11-26
> **Response to Reviewer 7hFZ (Part 1)**
>
> We would like to sincerely thank Reviewer 7hFZ for providing a detailed review and insightful questions. All the rebuttals have been included in the $\underline{\text{revised paper}}$.
>
> > **W1:** "The biggest weakness is that the proposed method is just substituting the immediate two steps that are done by traditional methods with data-driven methods. Thus, novelty is the biggest issue to me. Besides, the neural constitutive step is following the existing paper."
>
> Thanks for your detailed comments. However, we respectfully point out that NMP is more than a simple substitution of numerical modules, which is motivated by our thoughts on neural-physics integration and equipped with an elaborate modularization design. Here are the clarifications.
>
> **(1) Motivation: Our work is NOT just substituting two steps in traditional methods, but attempts to pursue a thorough modular architecture, which will bring unique advantages in scenario flexibility and efficiency over traditional simulators.**
>
> As stated in $\underline{\text{Lines 61-69 of Introduction}}$, this paper attempts to "design neural simulators with a thorough modular architecture, maintaining both data-driven flexibility and physical soundness", which is "far beyond simple substitution".
>
> With our special design in physics modularization and training strategy, NMP achieves:
>
> - **Better scenario flexibility:** As presented in $\underline{\text{Figure 5 of original submission}}$, under unknown frictional dynamics, both physics and neural-physics hybrid simulator fails, while NMP can still capture unknown physics from data and generate precise simulations.
> - **Better efficiency:** During the rebuttal, we further test the inference efficiency among physics, neural-physics simulators, and NMP, where NMP presents significantly faster speed, highlighting the efficiency benefit of thorough modularization. Here NCLaw's poor efficiency is because of its special SVD-based neural network design.
>
> | Bob Simulation Task           | Inference Time for 1000 steps (ms) |
> | - | - |
> | Traditional Simulator - step size 0.001         | 90275                              |
> | Neural-physics Hybrid (NCLaw) - step size 0.001 | 97121                        |
> | NMP (Ours)                    | 2250                               |
>
>
> **(2) Modularization design: Our design in substituting numerical modules is more than a thoughtless replacement, but with careful deliberation and elaborate designs.**
>
> As described in $\underline{\text{Lines 201-202 and 230 of Method section}}$, we define the modulization by replacing the key modules of the traditional simulator with neural networks respectively, which empowers NMP with favorable scenario flexibility.
>
> It is worth noticing that such a design is non-trivial, which stems from our analysis of the traditional computation pipeline ($\underline{\text{Section 3.1 of original submission}}$).
>
> **(3) Although we adopt the neural constitutive from NCLaw, NMP is under a fundamentally different computation pipeline.**
>
> We have discussed the relationship between NMP and NCLaw in $\underline{\text{Lines 208-210 of main text}}$. Since NMP is a thorough modular architecture and NCLaw is still a neural-physics hybrid simulator, these two models have several fundamental differences:
>
> - **Scenario flexibility:** Note that since NCLaw still contains a physics integration step, it suffers from the unknown physics scenario as presented in $\underline{\text{Figure 5 of original submission}}$, where the last row is NCLaw's simulation. This experiment highlights the difference in scenario flexibility between NMP and hybrid paradigms.
>
> - **Training stability:** As shown below, since NCLaw needs to connect neural module and physics integral during inference, it will suffer from training instability if the neural constitutive generates some physically incorrect intemediate results. In contrast, in NMP, the joint optimization of two modules enables better adaptation between two sub-process, bringing better long-term stability.
>
> | Bob task   | 1000-steps RMSE |
> | - | - |
> | NCLaw      | NAN             |
> | NMP (Ours) | 0.050           |
>
> Thus, without the entire modular pipeline, a single neural constitutive step could be meaningless in promoting scenario flexibility and training stability.
>
> We do hope the reviewer could reconsider our work's novelty from the perspective of the entire computation pipeline. We believe that the "novelty" of a work is not only about complicated architectural design or loss terms, but the contribution of insightful motivation and correct pipeline should not be overlooked.

---

> ### Author Response · Authors · 2025-11-26
> **Response to Reviewer 7hFZ (Part 2)**
>
> > **W2:** "Another contribution claimed by the authors is the separate training plus joint training. From what I understand, this training method was also proposed in the existing work to solve the collapse issue, which is also not new."
>
> Thank you for raising this point. While we agree that modular networks and separate training strategies have been explored in general-purpose machine learning, our contribution lies in **adapting and co-designing these ideas for physics simulation, which is an area where modularity remains underexplored**, as stated in $\underline{\text{Lines 134-135 of Related Work}}.$ Thus, we believe that the introduction and successful optimization of modular network in physics simulators is meaningful to this community.
>
> Besides, although "collapse" is a long-standing problem in modular networks, NMP's training strategy is specialized to the physics simulation context and co-designed with modularization architecture. **Without our modularization design in exposing internal force as the interface between two modules, we cannot leverage the intermediate physics quantity for separate training.**
>
> For clarity, we list the differences between our work and classical works in general modular networks as follows, which are different in the application domain, collapse definition and specific training strategy.
>
> |                      | Domain              | collapse problem definition            | Training strategy to avoid collapse            |
> | - | - | - | - |
> | Andreas et al., 2016 | Visual QA           | /                                      | specialized architecture                       |
> | Kirsch et al., 2018  | Language and Vision | homogeneous module selection           | data-dependent selection                       |
> | NMP (ours)           | Physics simulation  | meaningless interface physics quantity | supervision from intermediate physics quantity |
>
> We hope this clarifies that while modular training is not new in abstract terms, our work introduces a domain-specific adaptation that enables modular neural simulators in physical systems. We respectfully request the reviewer to consider this perspective on the contribution’s originality in the context of the entire simulation pipeline.
>
> > **W3:** "For the experiments, the authors compare it with purely data-driven methods. However, their method uses physics. I don’t think the experiments are good enough for showing the advantages of their methods. For example, they should compare with the traditional methods and we can see the gap when physics is used for both methods."
> >
> > **Q1:** "For strengthening the experiments, do the authors think the baselines can augment with physics knowledge?"
>
> During the rebuttal, we have made new experiments following your suggestion.
>
> **(1) Compare with traditional methods.**
>
> We already compare our approach to both pure physics-based methods and neural-physics hybrid methods  in $\underline{\text{Figure 5 of original submission}}$. Specifically, we evaluate on the *Friction* benchmark, which involves a deformable object sliding and coming to rest on a rough surface with unknown frictional dynamics.
>
> Our method is able to reproduce the observed frictional stopping behavior, unlike the pure physics simulator (which lacks this specific friction model) and NCLaw (which only learns the constitutive law but not contact dynamics). Our model successfully infers these hidden dynamics with friction.
>
> These results highlight the unique advantage of our method in scenarios with incomplete physical knowledge. While traditional simulators excel in fully specified environments, real-world systems often contain unknowns (e.g., contact, damping, or friction), where learned components become essential.
>
> **(2) Augment baselines with physics knowledge**
>
> As for pure neural baselines, since all of them are in the monolithic architecture, it is really hard to introduce "physics" to them; even their official papers do not consider "physics" integration. But following your suggestion, we have tried our best and found a possible way, where we make each model output three additional channels and supervise them with internal force.
>
> Introducing these three additional channls can inform the model with intermediate physics quantity (internal force) and help the whole simulation process. As shown below, NMP still achieves the best performance.
>
> | Bob Simulation Task           | 1000 steps RMSE |
> | - | - |
> | Transolver (2024) + Physics   | 1.44                |
> | EGNO (2024) + Physics         | 2.83                |
> | MeshGraphNet (2020) + Physics | 1.99                |
> | NMP (Ours)                    | 0.58                |

---

> ### Author Response · Authors · 2025-11-26
> **Response to Reviewer 7hFZ (Part 3)**
>
> > **W4:** "It will be good to see the advantages/disadvantages of the methods including baselines in terms of inference time."
>
> Following the reviewer's suggestion, we include the efficiency comparison as follows.
>
> | Bob Simulation Task  | Inference time for 1000 steps (ms) |
> | - | - |
> | Transolver (2024)     | 11796                              |
> | EGNO (2024)           | 26351                              |
> | MeshGraphNet (2020)   | 27603                              |
> | NCLaw (2023)          | 97121                       |
> | Traditional Simulator | 90275                              |
> | NMP (Ours)            | 2250                               |
>
> > **W5:** "The method is only tested on the specific problem, which means the scope is limited."
>
> First, we respectfully clarify that elastic simulation itself is a broad and fundamental domain, spanning applications in graphics, robotics, biomechanics, and materials science. By demonstrating generalization across multiple unseen conditions, geometries, and mesh resolutions, our method already covers a wide spectrum of elastic simulation tasks.
>
> Moreover, **even traditional simulators separate methods by domain: elastic simulation often uses the Finite Element Method (FEM), while fluid simulation typically relies on the Finite Volume Method (FVM) or grid-based schemes such as semi-Lagrangian advection and pressure projection**. As noted in $\underline{\text{Appendix D of original submission}}$, we acknowledge this domain specificity and view the extension of NMP to other physical domains as exciting future work.
>
> Still, we emphasize that NMP is a framework, not a simulator for elastic-only phenomena. Its modular design enables generalization to other domains by swapping in domain-appropriate physics modules and neural components. Below, we outline a feasible NMP extension for fluid simulation, drawing from classical simulation methods such as Stable Fluids [Stam 1999].
>
>
> | NMP extension to Stable Fluids [Stam 1999] | Design | Role
> | - | - | -
> | Module 1               |     Neural Advection Module   | Approximates advection of velocity fields or scalars; replaces semi-Lagrangian advection with a learned transport map.
> | Module 2               |  External Forcing Module      |  Applies learned body forces (e.g., buoyancy or control inputs), possibly conditioned on
> | Module 3               |    Neural Pressure Projection    | Solves for divergence-free condition (∇·u = 0); replaces Poisson solver with a learned correction operator enforcing incompressibility.
> | Module 4               |  Boundary Condition Module      |  Impose solid/wall and free‑surface boundaries.
>
> > **Q2:** "What is the error for predicting just one step forward for each method?"
>
> As per your request, we record the error among different methods in one step forward, where NMP still performs best.
>
> | Bob Simulation Task | 1-step RMSE | 1000-steps RMSE |
> | - | - | - |
> | Transolver (2024)   | 1.33e-3     | 1.782           |
> | EGNO (2024)         | 4.99e-3     | 2.770           |
> | MeshGraphNet (2020) | 3.51e-3     | 2.104           |
> | **NMP (Ours)**      | **6.61e-5** | 0.579           |
>
> > **Q3:** "For baselines, have you tried non-autoregressive prediction, e.g., directly predicting all the following 100 steps."
>
> Thanks for pointing out this new experimental setting.
>
> First, we want to highlight that step-by-step autoregressive prediction is the official configuration in all three neural baselines, that is why we compare them in the autoregressive way.
>
> Following your suggestion, we retrained all three neural simulator baselines to directly predict future 100 steps and rollout 10 times for the future 1000 steps. As shown below, although directly predicting 100 steps can be helpful in reducing the accumulation error, these baselines still fall behind NMP.
>
> | Bob Simulation Task                            | 1000-steps RMSE |
> | - | - |
> | Transolver (2024) + direct predict 100 steps   | 0.75            |
> | EGNO (2024) + direct predict 100 steps         | 2.00            |
> | MeshGraphNet (2020) + direct predict 100 steps | 2.61            |
> | **NMP (Ours) + autoregressive prediction**     | **0.58**        |

---

### Author Response · Authors · 2025-11-26
**Summary of Revisions**

Dear Reviewers,

We sincerely thank all the reviewers for their valuable feedback, which has been instructive in helping us improve our paper. And also, thanks for your patience in waiting for the rebuttal. For every detailed review, we would like to include it, along with the corresponding revision, in our paper, which requires some time in setting up new experimental scenarios, training models, etc.

Our original submission has generally received positive feedback from reviewers, in that our work is "a **fairly new** approach", "**a novel result** in the neural simulation literature", "an **excellent proof-of-concept**" and "**addresses a key point of neural simulators**"; the paper includes "**good empirical evidence**" and is "**easy to follow and well organized**".

The reviewers also raised insightful questions. After two weeks' effort, we have tried our best to resolve all the reviewers' questions. To sum up, we have conducted **more than 30 new experiments** and set up a **new "sim to real" scenario** from scratch. Here is a summary of our rebuttal:

- **Efficiency comparison among different methods (Reviewers 7hFZ, ZLNW, b4yZ):** We have recorded all the baseline efficiencies, where NMP is significantly superior to traditional methods (around 40x faster) and is 5x faster than advanced neural simulators. Comparison has been included in $\underline{\text{Table 2 of revised paper}}$.
- **NMP's advantage w.r.t. traditional simulators (Reviewers 7hFZ, BcK5):** By conducting extensive new experiments, we have demonstrated that NMP has advantages in (1) scenario flexibility, (2) efficiency and (3) transferability over traditional simulators. For clarity, we also rephrase $\underline{\text{Section 4.2}}$ as $\underline{\text{comparison with physics simulators}}$.
- **NMP's application in real-world simulations (Reviewers BcK5, b4yZ):** To address the reviewers' concern, we newly set up a cube-real-world scenario with an unknown external force, where only position information is accessible. Taking advantage of the transferability of neural networks, NMP can successfully simulate dynamics in this partially observable scenario, posing the potential of NMP in real-world applications. The experiment has been included in $\underline{\text{Figure 6 of revised paper}}$.
- **NMP's extension to other domains (Reviewers 7hFZ, ZLNW):** As mentioned in the limitation part of the original submission, the idea of neural modular physics is general. After an in-depth investigation, we propose a possible way to extend NMP to fluid simulation, whose modularization design follows a classical paper [Stam 1999]. This discussion has been included in the $\underline{\text{Conclusion of revised paper}}$.
- **Compare with enhanced baselines (Reviewer 7hFZ):** Following the reviewer's suggestion, we enhanced the baselines by adding physics supervision and enlarging the prediction length at each step. Still, NMP significantly outperforms these baselines in long-term simulation, highlighting the advancement of our design.
- **Discussion about rigid transformation (Reviewer BcK5):** We clarify the detailed design of our model and demonstrate that NMP can exactly guarantee zero stress in rigid transformation.
- **Hyperparameter analysis on physical constraint (Reviewer ZLNW):** Just like the reviewer mentioned, too large a physical constraint will overwhelm the optimization process. See $\underline{\text{Appendix D of revised paper}}$ for more discussions.
- **Compute physics validation metrics (Reviewer b4yZ):** Thanks for this great suggestion. We newly measured the local volume preservation as the physics metric, where NMP is 2x better than other baselines in this physical metric.
- **Failure mode analysis:** Following the reviewer's suggestion, we select the worst case of NMP and provide a visual comparison with other models in $\underline{\text{Appendix E of revised paper}}$. Although large deformation is challenging for NMP, our method still performs best.
- **Rephrase writing issues (Reviewers ZLNW, b4yZ):** We deeply thank these detailed reviews and have resolved all the typos in the $\underline{\text{revised paper}}$.

All the above rebuttals have been included in the $\underline{\text{revised paper}}$ and highlighted in blue. Hope these new experiments and revised paper can resolve all your concerns to your satisfaction.

We thank you again for your time and dedication.

---

### Meta-Review · Area_Chair_cpua · 2026-01-06

**Summary:**

This paper proposes a simulation framework that combines the efficiency of neural networks with the modularity characteristic of classical numerical methods. In particular, such modularity is typical of the finite element method (FEM), and the proposed approach aims to improve simulation efficiency by replacing the FEM constitutive equations and time evolution with neural networks. The paper details the method specifically designed for elastic problems, and experiments under various conditions are shown. While reviewers acknowledge the importance of this approach, some criticisms, such as its advantages over FEM and the narrow scope, have been raised. The authors address these points to some extent, but in my opinion, it seems to be slightly below the borderline. For this paper to be accepted, it would be desirable to extend it to a broader range of applications, such as fluid dynamics, as the authors mentioned as future work.

**Reviewer Concerns:**

The main concerns are whether it is merely a simple replacement for FEM, its advantages over FEM, and the limited scope. In response, the authors argue that modularity is a very important contribution and that there are problems that cannot be handled by FEM. These are certainly true, but the illustrations of problems that cannot be handled by FEM are limited. Furthermore, considering that modularity can be generalized beyond elastic problems, there remains room for improvement. Therefore, the reviewers' concerns may not be fully addressed.

**Reviewer Scores:**

The scores were 8-6-4-4. The most positive reviewer acknowledged that his/her concerns are addressed.  To some extent, the score might have improved, and in that case, it might have become 8-6-6-4. This possible result is not a low score, but it seems to be slightly below the borderline.

---

### Decision · Program_Chairs · 2026-01-26

Reject